biomathematics/systems biology/bioengineering

entrainment, contractive systems, systems biology, gene expression, ribosome flow model, optimal control theory

**Author for correspondence:**
Eduardo D. Sontag
e-mail: sontag@sontaglab.org

# Maximizing average throughput in oscillatory biochemical synthesis systems: an optimal control approach

M. Ali Al-Radhawi[1], Michael Margaliot[2] and Eduardo D. Sontag[1,3]

[1]Departments of Bioengineering and Electrical and Computer Engineering, Northeastern University, Boston, MA 02115, USA
[2]Department of Electrical Engineering-Systems, Tel Aviv University, Tel Aviv, Israel 69978
[3]Laboratory of Systems Pharmacology, Program in Therapeutic Science, Harvard Medical School, Boston, MA 02115, USA

  MM, 0000-0001-8319-8996; EDS, 0000-0001-8020-5783

A dynamical system *entrains* to a periodic input if its state converges globally to an attractor with the same period. In particular, for a constant input, the state converges to a unique equilibrium point for any initial condition. We consider the problem of maximizing a weighted average of the system's output along the periodic attractor. The *gain of entrainment* is the benefit achieved by using a non-constant periodic input relative to a constant input with the same time average. Such a problem amounts to optimal allocation of resources in a periodic manner. We formulate this problem as a periodic optimal control problem, which can be analysed by means of the Pontryagin maximum principle or solved numerically via powerful software packages. We then apply our framework to a class of nonlinear occupancy models that appear frequently in biological synthesis systems and other applications. We show that, perhaps surprisingly, constant inputs are optimal for various architectures. This suggests that the presence of non-constant periodic signals, which frequently appear in biological occupancy systems, is a signature of an underlying *time-varying* objective functional being optimized.

## 1. Introduction

Periodic oscillations are abundant in biomolecular systems, and an extensive body of research has been devoted to study their roles in

intracellular and extracellular interactions [1,2]. In the presence of such excitations, proper functioning of biological systems often requires their internal states to *synchronize* with the periodic input signal. In the parlance of systems theory, this is known as *entrainment*, which means that the response of a system subject to a periodic input with period $T$ will converge to a periodic trajectory of the same period $T$. There has been great recent interest in the study of this phenomenon [3–7]. Examples of external periodic influences include operation under the influence of sunlight, which requires the internal clocks of biological organisms to entrain to the 24-hour solar day. For instance, it has been shown that the plant *Arabidopsis* uses its circadian clock to anticipate times with an increased susceptibility to fungal pathogens and regulates its immune system resources accordingly [8]. Entrainment is also essential in many *synthetic* biological systems. For instance, synthetic oscillators can be used to emulate natural hormone release rhythms in the treatment of certain diseases [9]. More generally, robust and optimal synthetic oscillators constitute an important module in larger systems [10,11].

At the intracellular level, the *cell cycle* is a periodic routine that regulates DNA replication and cell division. This requires precise regulation of many interacting proteins and also appropriate resource allocation at different stages of the cell cycle. Deviations from the programme can lead to cell death or cancer.

An important underlying process is translation, which is a major component in the central dogma of molecular biology, and requires sophisticated coordination among ribosomes, mRNA and tRNA molecules, and various proteins. Two of the key underlying steps are *initiation* in which the ribosome attaches to an mRNA molecule, and *elongation*, in which the ribosome scans along the mRNA to produce a chain of amino acids. Regulation of initiation and elongation are an effective way to control protein concentrations [12,13].

One biological mechanism for cell cycle-regulated genes is based on codons whose corresponding tRNAs have low abundances (known as non-optimal codons) [14,15]. In particular, *periodic* variations in the level of these specific tRNAs can generate cell cycle-dependent oscillations in the corresponding protein levels [14]. In other words, the protein levels entrain to the periodic excitation provided by the tRNA levels. Similar oscillation-inducing regulation mechanisms during DNA damage response have also been reported [16]. Other works have indicated that the speed of translation is sensitive to fluctuating tRNA availability [17], that cells use tRNA to control protein abundance in stress conditions [18], and that tRNA dysregulation is a contributing factor in cancer progression [19]. In addition to tRNA regulation, many other intracellular oscillators have been identified as regulators of the cell cycle [20,21].

Here, we analytically investigate the hypothesis that periodic rates in the cellular environment are used to maximize gene expression throughput. To that end, we first describe, as a motivation, a class of mathematical models that are useful in modelling various processes involved in gene translation. Our focus is to analyse these models in the presence of periodic excitations modelled as periodic inputs that attempt to maximize a certain reward function that is proportional to the throughput of the system. We pose these problems in the rigorous language of optimal control theory and analyse them under a variety of assumptions.

## 1.1. Motivation: occupancy models

In many important biological models, state variables describe the occupancy in a certain site or a compartment. For example, in physiology, compartmental models describe drug absorption distribution and elimination in various body fluids or tissues [22].

### 1.1.1. A one-dimensional model

Many biological processes involve 'biological machines' that move along a one-dimensional lattice of ordered 'sites'. Examples include ribosomes that scan the mRNA molecule during translation, molecular motors that carry cargoes along a filamentous network in the cytoskeleton, and phosphotransferases that transfer the phosphoryl group from the sensor kinases to some ultimate target. To be concrete, we focus on ribosomes and mRNA translation, but the same ideas apply to other models.

We now derive such a one-dimensional occupancy model using several alternative modelling approaches. Let $X$ ($Z$) be the species denoting bound (unbound) ribosomes. The free ribosomes bind to mRNA. Bound ribosomes need tRNAs to translate the information in the mRNA into proteins ($P$). A phenomenological one-step model written in chemical reaction network (CRN) formalism [23] gives

$$mRNA + Z \rightarrow X$$
$$tRNA + X \rightarrow Z + mRNA + P.$$

We assume that tRNA and mRNA are abundant, so that their dynamics are not affected by the aforementioned reactions. Note that the species tRNA represents all possible variants of transfer RNA. Let $x(t)$ be the concentration of occupied (bound) ribosomes in the cell at time $t$, and let $z(t)$ be the concentration of free ribosomes. The occupancy of ribosomes is determined by mRNA transcript abundance $u_0(t)$ and tRNA abundance $u_1(t)$. The CRN gives the following system of bilinear ODEs:

$$\left.\begin{array}{l} \dot{x}(t) = u_0(t)z(t) - u_1(t)x(t), \\ \dot{z}(t) = u_1(t)x(t) - u_0(t)z(t). \end{array}\right\} \tag{1.1}$$

Assuming a fixed total concentration of ribosomes $M$, we have $x(t) + z(t) \equiv M$. The total concentration can be normalized to $M = 1$. Then the two-dimensional dynamics can be reduced to a one-dimensional ODE

$$\dot{x}(t) = u_0(t)(1 - x(t)) - u_1(t)x(t). \tag{1.2}$$

This implies that $x(t)$ evolves on the unit interval, and it can be interpreted as a normalized occupancy of some site at time $t$. More generally, $x(t)$ can be interpreted as the probability that a certain site is occupied by some 'biological machine', like a ribosome or a molecular motor, at time $t$. This occupancy model has been termed a 'bottleneck' module in [24].

Note that the occupancy model (1.1) can also be used to model binding and unbinding of a substrate to an enzyme.

### 1.1.2. Multisite models: the ribosome flow model

The Totally Asymmetric Simple Exclusion Process (TASEP) [25] is a fundamental stochastic model from non-equilibrium statistical physics. In TASEP, particles move forward at random times along a one-dimensional chain of sites. A site can be either free or contain a single particle. Totally asymmetric means that the flow is unidirectional, and simple exclusion means that a particle can only hop into a free site. This models the fact that two particles cannot be in the same place at the same time. The simple exclusion paradigm generates an indirect coupling between the particles and also allows modelling the evolution of 'traffic jams': if a particle remains at site $i$ for a long time, then particles will accumulate 'behind' it, i.e. in site $i - 1$, then site $i - 2$ and so on. TASEP has been used extensively to model and analyse ribosome flow [26] and many more natural and artificial processes including molecular motors, traffic flow, evacuation dynamics and more [27].

The *ribosome flow model* (RFM) [28] is the dynamic mean-field approximation of TASEP. In the RFM, the state variables $x_1(t), \ldots, x_n(t)$ describe the occupancy in $n$ sites along the mRNA molecule. The RFM dynamics is described by a system of $n$ first-order ODEs

$$\dot{x}_k = \lambda_{k-1}x_{k-1}(1 - x_k) - \lambda_k x_k(1 - x_{k+1}), \quad k = 1, \ldots, n, \tag{1.3}$$

where we define $x_0(t) \equiv 1$ and $x_{n+1}(t) \equiv 0$. Here, $x_i(t)$ describes the occupancy at site $i$ at time $t$, normalized such that $x_i(t) = 0$ [$x_i(t) = 1$] means that site $i$ is completely empty [full] at time $t$. In the context of translation, $\lambda_i(t) > 0$ describes the transition rate from site $i$ to site $i + 1$ at time $t$. This rate depends on various biomechanical properties, for example, the abundance of tRNA molecules delivering the amino acids to the ribosomes. Equation (1.3) can be explained as follows. The change in the density in site $k$ is the flow from site $k - 1$ into site $k$ minus the flow from site $k$ to site $k + 1$. The first term, $\lambda_{k-1} x_{k-1} (1 - x_k)$, is proportional to the transition rate from site $k - 1$ to $k$, the occupancy at site $k - 1$, and the amount of 'free space' $(1 - x_k)$ at site $k$. Note that this is a 'soft' version of simple exclusion: as site $k$ fills up, the flow into it decreases. The second term is similar. Note that $\lambda_n(t)x_n(t)$ describes the flow of ribosomes out of the last site at time $t$, i.e. the protein production rate. If the whole mRNA strand is considered as one site, that is, $n = 1$, then the RFM model will be identical with the occupancy model (1.2).

The state space of the RFM is the $n$-dimensional unit cube $[0, 1]^n$. It was shown in [29] that the RFM (with constant $\lambda_i$'s) admits a unique equilibrium $x_e = x_e(\lambda_0, \ldots, \lambda_n) \in (0, 1)^n$ and that for any $a \in [0, 1]^n$, the corresponding solution of (1.3) satisfies $\lim_{t \to \infty} x(t; a) = x_e$. In other words, the transition rates determine a unique globally asymptotically stable (GAS) equilibrium. More generally, [6] showed that if the rates are time varying, and jointly periodic with a period $T$, then (1.3) admits a unique solution $\gamma_T : \mathbb{R}_+ \to (0, 1)^n$, i.e. $T$-periodic, and $x(t, a)$ converges to $\gamma_T$ for all $a \in [0, 1]^n$. In other words, the RFM entrains. Note that a constant rate is $T$-periodic for any $T$, so entrainment also holds if a single rate is $T$-periodic and all the other rates are constant. In the biological context, entrainment can be interpreted as follows: if, say, variations in tRNA abundances generate $T$-periodic initiation and/or elongation rates, then the protein production rate will also converge to a periodic pattern with period $T$.

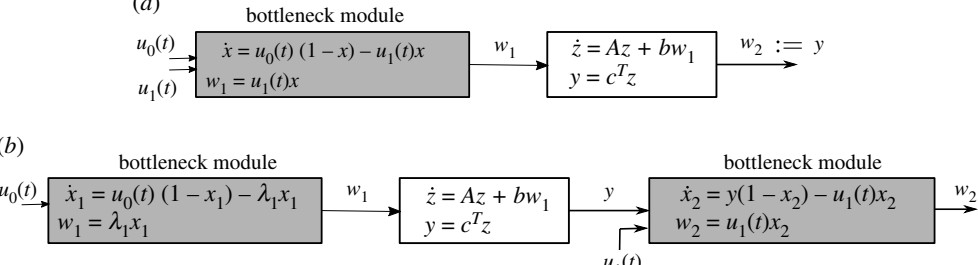

**Figure 1.** Two examples of generalized occupancy models. The controls are $u_0(t)$, $u_1(t)$ which are scalar functions. We have $x_1$, $x_2$, $x$, $w_1$, $w_2$, $y \in \mathbb{R}_+$, $z \in \mathbb{R}^n$, $A \in \mathbb{R}^{n \times n}$, $b$, $c \in \mathbb{R}^n_+$. The linear system block is assumed to be positive and Hurwitz.

The RFM and its variants have been used extensively to model and analyse ribosome flow during the process of translation (see e.g. [30–34]), as well as other important cellular processes like phosphorelay [35].

Just like TASEP, the RFM (and in particular the model (1.1)) is a phenomenological model that can be applied to study various processes like vehicular or pedestrian traffic [24]. In this case, the occupancy is interpreted as the ratio between the number of vehicles (or pedestrians) at a certain junction at time $t$ and the total number of possible vehicles.

### 1.1.3. Generalized occupancy models

Let $\mathbb{R}^n_+ := \{ x \in \mathbb{R}^n \, | \, x_i \geq 0, \; i = 1, \ldots, n \}$ denote the non-negative orthant in $\mathbb{R}^n$. For two vectors $a, b \in \mathbb{R}^n$, we write $a \geq b$ if $a_i \geq b_i$ for all $i = 1, \ldots, n$. Recall that the linear single-input single-output (SISO) linear system

$$\dot{x} = Ax + bu$$

and

$$y = c^T x$$

is called *positive* if every entry of $b$, $c$, and every off-diagonal entry of $A$ is non-negative (i.e. $A$ is a *Metzler* matrix). This implies that for any $x(0) \geq 0$ and any control $u$ with $u(t) \geq 0$ for all $t$ we have $x(t) \geq 0$ and $y(t) \geq 0$ for all $t \geq 0$ [36]. This is useful when the state variables and the output represent physical quantities that can never attain negative values, e.g. population sizes or concentrations of molecules.

*Generalized occupancy models* (GOMs) are a cascade of occupancy models and SISO positive linear systems. These models are useful when the output of an occupancy model is the input to another biological system that, in the vicinity of its equilibrium point, can be approximated as a positive linear dynamical system. Similar to the multi-site RFM model introduced before, it can be shown that GOMs entrain to periodic inputs.

For example, figure 1a depicts a time-varying bottleneck module feeding a positive linear system. In this module, $u_0$, $u_1$ are entrance rates, and $w_1$ is the exit rate. The effective inflow is proportional to the *vacancy* $1 - x(t)$, while the outflow is proportional to the *occupancy* $x(t)$. This cascade models an occupancy model driving a downstream linear system. As another example, figure 1b depicts a linear system 'sandwiched' between a two-site RFM and one-site RFM. This can model a situation where the production rate of one protein affects, via another biological process, the promoter (and thus the initiation rate) of some other mRNA.

A GOM can also be used to model the RFM with time-varying rates under the condition

$$\lambda_i(t) \gg \lambda_0(t) \quad \text{for all } i \geq 1 \text{ and all } t \geq 0. \tag{1.4}$$

Then, we can expect that the initiation rate becomes the bottleneck rate and thus $x_i(t)$, $i = 2, \ldots, n$, converge to values that are close to zero, suggesting that (1.3) can be simplified to

and
$$\left. \begin{aligned} \dot{x}_1(t) &= \lambda_0(t)(1 - x_1(t)) - \lambda_1(t)x_1(t) \\ \dot{x}_i(t) &= \lambda_{i-1}(t)x_{i-1}(t) - \lambda_i(t)x_i, \;\; i \in \{2, \ldots, n\}, \end{aligned} \right\} \tag{1.5}$$

which has the same form as the cascade in figure 1a.

After these motivating examples, we next formulate the abstract questions to be studied in this article.

## 1.2. Gain of entrainment

To analyse the effect of periodic signals on the performance of an occupancy system, we focus on a class of systems whose internal variables *entrain*, i.e. follow the rhythms of an external driving signal. Entrainment can be studied in the framework of systems and control theory. The periodic excitation is modelled as the control input $u(t)$ of a dynamical system, and the system entrains if in response to a $T$-periodic excitation it admits a globally attractive $T$-periodic solution $\gamma_T$. In other words, every solution of the system converges to the attractor $\gamma_T$.

Assuming that a system entrains, the next question is: what is the *quantitative* potential advantage of entrainment? In other words, is entrainment to periodic inputs more advantageous when optimizing a certain objective function? If the answer is affirmative, then we say that the system exhibits a positive *gain of entrainment*. To explain this, consider a control system that, for any $T \geq 0$ and any $T$-periodic control $u_T$, entrains to a unique $T$-periodic solution $\gamma_T$. Note that, in particular, this implies that for any constant control $u(t) \equiv u_0$ the trajectory converges to a unique equilibrium $\gamma_0$ for any initial condition. Suppose also that the system admits a scalar output $y(t) = h(t, x(t), u(t))$ (i.e. a function of time, state and the input), and that $h$ is $T$-periodic in $t$, so that the output also entrains. The output represents a quantity that we would like to maximize, e.g. traffic flow or protein production rate.

Since the system entrains, we ignore the transients and consider the problem of maximizing the average of the periodic output, that is, the average over a period of $h(t, \gamma_T, u_T)$. The gain of entrainment is the benefit (if any) in the maximization for a (non-trivial) periodic control over a constant control. A natural example is to analyse the gain in traffic flow for periodically varying traffic signals over constant signals. However, to make this meaningful, we must add another assumption, namely, that the total time of green lights in both alternatives is equal. Mathematically, this means that we compare the average output for a time-periodic control $u_T$ and a constant control $\bar{u}$ such that the average value of $u_T$ over a period is equal to $\bar{u}$. If the gain of entrainment is positive, then entrainment does not only assist in producing an internal clock that can follow an external periodic excitation but also yields higher production rates than those obtained by equivalent constant excitations.

The possible advantages of periodic forcing of various production processes are well known. For example, [37] states that: '…theoretical and experimental studies have shown that the performance (for instance micro-algae or bio-gas production) of some optimal steady-state continuous bioreactors can be improved by a periodic modulation of an input such as dilution rate or air flow'. Reference [38] studies a partial differential equation (PDE) model for harvesting a biological resource and demonstrates the advantages of periodic harvesting over a constant one.

The gain of entrainment was recently introduced in [24]. Entrainment in nonlinear systems is non-trivial to prove. A typical proof is based on contraction theory [6,7], yet this type of proof provides no information on the attractive periodic solution, except for its period (see [39] for some related considerations). Nevertheless, we show here that determining the gain of entrainment can be cast as an *optimal control problem*. This allows using powerful theoretical tools, like Pontryagin's maximum principle [40–42], as well as numerical methods in studying the gain of entrainment. We demonstrate this by analysing the gain of entrainment in several examples of occupancy models.

For instance, consider the gain of entrainment for (1.5). It is natural to speculate that using time-periodic rates $\lambda_i(t)$, which are properly synchronized, yields a positive gain of entrainment with respect to using constant rates (with the same average values). In the context of traffic flow, this is equivalent to the conjecture that properly synchronized periodic traffic lights can improve the overall flow. However, we show that, perhaps surprisingly, for a subclass of these systems, the gain of entrainment is in fact zero.

We also consider a problem formalism that allows for time-varying costs of resources, like tRNAs, along the period. These may be produced at different unit costs at different times of the cycle. This modified formulation allows the allocation of resources differently at different times along the cycle. Also, instead of average throughput, a *weighted* average of the product may be more relevant, in the sense that we may need certain enzymes at different times of the day or at different points in the cell cycle. This corresponds to 'just-in-time production' [43]. In such cases, we show, not so surprisingly, that time-varying periodic inputs may indeed offer an advantage over constant inputs. This suggests

that the presence of non-constant periodic signals, which frequently appear in biological occupancy systems, implies that the system is optimizing an underlying *time-varying* objective functional.

Our work is related to results from the field of optimal periodic control (OPC) (e.g. [44]). As noted by Gilbert [45], OPC was motivated by the following question: does time-dependent periodic control yield better process performance than optimal steady-state control? In particular, the recent paper [46] defines a notion called *over-yielding* that is closely related to the gain of entrainment. However, our setting is different, as in OPC periodicity was enforced by restricting attention to controls $u$ guaranteeing that $x(T) = x(0)$. This implies in particular that the initial value $x(0)$ (and thus also general transient behaviours) may have a strong effect on the results. Also, in the typical OPC formulation, there is in general no requirement that the averages of the periodic and constant controls are equal.

We study systems that entrain, and thus for a $T$-periodic control, the state of the system converges to a unique $T$-periodic trajectory for *any* initial condition $x(0)$. In other words, we consider the behaviour of attractors.

The remainder of this article is organized as follows. The next section defines the gain of entrainment for a general mathematical model. Section 3 shows how the analysis can be cast as an optimal control problem. Section 4 demonstrates the theory for the two-input bottleneck module. Section 5 proves that for several GOMs, including the ones depicted in figure 1, the gain of entrainment is zero. Finally, conclusions and future directions are presented in §6. Appendix A contains proofs of the results including a detailed analysis characterizing extremals via the Pontryagin maximum principle (PMP).

# 2. Gain of entrainment

We consider a general nonlinear control system

$$
\begin{rcases}
\dot{x} = f(x, u) \\
y = h(t, x, u),
\end{rcases}
\tag{2.1}
$$

and

with $f$, $h$ locally Lipschitz functions, state $x(t) \in \mathbb{R}^n$, control (or input) $u(t) \in \mathbb{R}^m$, and scalar output $y(t) \in \mathbb{R}$. We allow $h$ to be time varying to include the cases in which different weights can be used at different times in the cycle. The set of admissible controls consists of measurable functions taking values in some compact set $U \subset \mathbb{R}^m$. Let $x(t, p, u)$ denote the solution of (2.1) at time $t \geq 0$ for the initial condition $x(0) = p$ and the control $u$. We assume throughout that for any $x(0)$ in the state space and any admissible control, (2.1) admits a unique solution for all $t \geq 0$.

We say that system (2.1) *entrains* if in response to any admissible and $T$-periodic control $u_T(t)$ the system admits a unique $T$-periodic solution $\gamma_T(t)$ (that depends on $u_T$), and for any initial condition $p$, the solution $x(t, p, u_T)$ converges to $\gamma_T$. This implies in particular that the system 'forgets' its initial condition.

To explain the mathematical formulation of the gain of entrainment, fix $q \in \mathbb{R}^m$ with $q_i > 0$ for all $i$. We would like to consider only inputs whose average over a period is $q$ and compare their effect to the effect of the constant control $u(t) \equiv q$. However, we allow a slightly more general scenario by fixing a weighting function $\alpha(t) > 0$ such that $\frac{1}{T} \int_0^T \alpha(t) \, \mathrm{d}t = 1$. We then restrict attention to $T$-periodic controls satisfying the weighted integral constraint

$$
\frac{1}{T} \int_0^T \alpha(t) u(t) \, \mathrm{d}t = q,
\tag{2.2}
$$

that is, the $\alpha$-weighted average of $u$ is $q$. This can be further generalized by allowing a general measure $\mu$ on the interval $[0, T]$ and imposing $\int_{[0,T]} u(t) \, \mathrm{d}\mu = q$. However, we keep the presentation simple by adhering to (2.2).

Let

$$
z(u) := \frac{1}{T} \int_0^T h(t, \gamma_T(t), u(t)) \, \mathrm{d}t,
\tag{2.3}
$$

that is, the average value of the output along the globally attractive $T$-periodic solution (recall that we assume that $h$ is $T$-periodic in its first variable). If the convergence to $\gamma_T$ is relatively fast, then after a short transient the average output over a period of length $T$ is very close to $z(u)$. In applications in

fields like biotechnology and traffic control, the average value of the output, and not its specific values at all times, is often the relevant quantity.

The constant control $u(t) \equiv q$, which we simply denote by $q$, is also $T$-periodic (for any $T \geq 0$) and satisfies (2.2). Hence, the corresponding solution converges to a fixed point $e = e(q)$ and

$$z(q) = \frac{1}{T}\int_0^T h(t, e, q) \, dt$$
$$= h(e, q).$$

The *gain of entrainment* of (2.1) is defined as follows:

$$c_T(q) := \sup_u z(u) - z(q), \tag{2.4}$$

where the sup is over all admissible, $T$-periodic controls that satisfy the constraint (2.2). Thus, we are always comparing the effect of controls with the same average value. Note that $c_T(q) \geq 0$ for all $q$. If $c_T(q) > 0$ for some $q$, then there exists a non-trivial periodic control that yields a higher average output than that obtained for a constant control. If $c_T(q) = 0$, then non-trivial $T$-periodic controls are 'no better' than the simple constant control equal to $q$.

To gain a wider perspective, consider the case of a SISO asymptotically stable linear time-invariant (LTI) system with input [output] $u(t)$ [$y(t)$] and transfer function $G(s)$. Fix $T > 0$, and consider the $T$-periodic control

$$u_T(t) := a + b\sin\left(\frac{2\pi t}{T}\right),$$

with $a, b \in \mathbb{R}$. Note that $\frac{1}{T}\int_0^T u_T(t) \, dt = a$. Let $\omega := 2\pi/T$. It is well known that the output converges to the $T$-periodic function $y_T(t) := G(0)a + |G(j\omega)|b\sin(\omega t + \angle G(j\omega))$, where $j := \sqrt{-1}$, so $(1/T)\int_0^T y_T(t) \, dt = G(0)a$. On the other hand, for the constant control $u(t) \equiv a$, the output converges to $G(0)a$, which is the same value. Thus, for this input, the gain of entrainment is zero. Any $T$-periodic, measurable, and bounded input can be expressed as a Fourier series in terms of sinusoidal functions, and this implies that for LTI systems, the gain of entrainment is always zero.

However, for nonlinear system, the gain of entrainment may be positive. The next two examples demonstrate this.

**Example 2.1.** Consider the scalar system:

and
$$\left.\begin{array}{l} \dot{x}(t) = 1 - x(t)u(t) \\ y(t) = x(t). \end{array}\right\} \tag{2.5}$$

Fix $T > 0$. For a function $v : \mathbb{R}_+ \to \mathbb{R}_+$, let $\bar{v} := (1/T)\int_0^T v(s) \, ds$. Fix $q > 0$. For the control $u(t) \equiv q$ any solution of (2.5) converges to the equilibrium $q^{-1}$. Consider a $T$-periodic and positive control $u_T(t)$ satisfying $\bar{u}_T = q$, and assume there exists some $\alpha > 0$ such that $u_T(t) \geq \alpha$ for almost all $t \in [0, T]$. Then any matrix measure of the Jacobian of (2.5) is uniformly less than or equal to $-\alpha < 0$. Therefore, the system is contractive and any solution of (2.5) converges to a unique $T$-periodic solution $x_T(t)$.

Let $\omega := 2\pi/T$. Consider now the specific $T$-periodic control

$$u_T(t) := 1 + \left(\frac{1}{2}\right)\cos(\omega t). \tag{2.6}$$

Here, $q = \frac{1}{T}\int_0^T u_T(t) \, dt = 1$. For this input, the corresponding solution of (2.5) is

$$x(t) = \exp\left(-t - \frac{\sin(\omega t)}{2\omega}\right)(x(0) + \phi(t)),$$

where $\phi(t) := \int_0^t \exp(s + (\sin(\omega s)/2\omega)) \, ds$. In particular,

$$x(T) = \exp(-T)(x(0) + \phi(T)).$$

The initial condition $x(0) = c$ for which the solution is $T$-periodic is

$$c = \exp(-T)(c + \phi(T)),$$

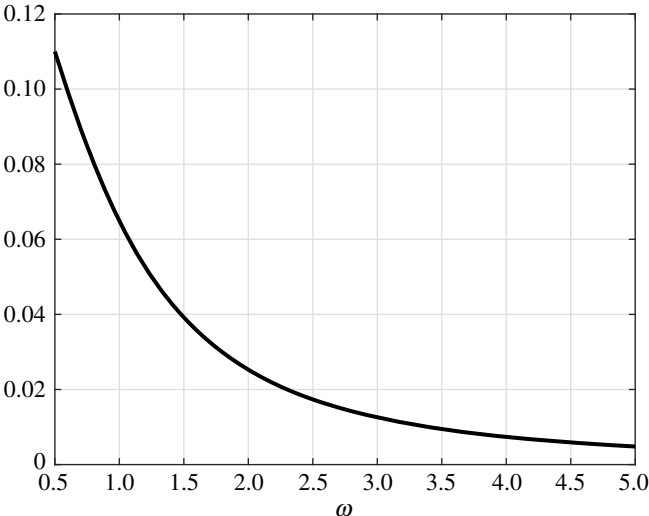

**Figure 2.** Average value of $x_T(t) - 1$ in example 2.1 as a function of $\omega \in [0.5, 5]$.

so

$$c = \frac{\exp(-T)\phi(T)}{1 - \exp(-T)}. \tag{2.7}$$

Thus, the attractive periodic solution is $x_T(t) := \exp\left(-t - (\sin(\omega t)/2\omega)\right)(c + \phi(t))$. The average of the control $u_T(t)$ is $q = 1$. On the other hand, for the control $u_0(t) \equiv 1$, the solution of (2.5) converges to the steady-state 1. Figure 2 depicts the value

$$\frac{1}{T}\int_0^T x_T(t)\, \mathrm{d}t - 1,$$

as a function of $\omega = 2\pi/T$. It may be seen that this is always positive and is maximal as $T \to \infty$.

We conclude that for $q = 1$ the gain of entrainment of (2.5) is positive for any $T > 0$. Note that for large values of $\omega$, the gain of entrainment goes to zero. This is expected due to *averaging* [47]. Roughly speaking, for large values of $\omega$, the system cannot track the fast changes in the input and thus responds to the average of the input. More rigorously, for a system affine in the control, the map from controls on an interval $[0, T]$ to trajectories on $[0, T]$ is continuous with respect to the weak* topology in $L^1$ and the uniform topology on continuous functions, respectively (e.g. [48, theorem 1]), and for a periodic input $u(t)$, the input $u(\omega t)$ converges weakly to the average of $u$. An alternative proof is given for example in the textbook [47] (Section 10.2) (changing time scale in the statement of theorem 10.4, by $x(t) = x(t/\varepsilon)$).

**Example 2.2.** Consider the system

$$\left.\begin{aligned}
\dot{x}_1(t) &= -x_1(t) + u(t), \\
\dot{x}_2(t) &= -x_2(t) + ax_1^2(t) \\
y(t) &= x_2(t),
\end{aligned}\right\} \tag{2.8}$$

and

with $a > 0$.

Consider the input $u_T(t) := 1 + \sin(\omega t)$, with $\omega > 0$. Here, $q = \bar{u}_T = 1$. Let $T := 2\pi/\omega$, and let $H(s) := 1/(1 + s)$. Then, $x_1$ converges to the steady-state solution

$$x_{1T}(t) := 1 + |H(j\omega)|\sin(\omega t + \angle H(j\omega)).$$

Hence,

$$x_{1T}^2(t) = 1 + 2|H(j\omega)|\sin(\omega t + \angle H(j\omega)) + |H(j\omega)|^2\sin^2(\omega t + \angle H(j\omega))$$

$$= 1 + \frac{|H(j\omega)|^2}{2} + 2|H(j\omega)|\sin(\omega t + \angle H(j\omega)) - \frac{|H(j\omega)|^2}{2}\cos(2\omega t + 2\angle H(j\omega)).$$

It follows that $x_2$ converges to the steady-state solution

$$x_{2T}(t) := a\left(1 + \frac{|H(j\omega)|^2}{2} + 2|H(j\omega)|^2 \sin(\omega t + 2\angle H(j\omega))\right.$$
$$\left. - \frac{|H(j\omega)|^2}{2}|H(2\omega)|\cos(2\omega t + 2\angle H(jw) + \angle H(2j\omega))\right),$$

so

$$\bar{x}_{2T} = \frac{1}{T}\int_0^T x_{2T}(t)\,\mathrm{d}t = a\left(1 + \frac{|H(j\omega)|^2}{2}\right).$$

On the other hand, for the average input $\bar{u}_T = 1$, $x_1(t)$ converges to one, and $x_2(t)$ to $a$, so the average of the output is $a$. The difference between the two averaged outputs is thus

$$\frac{a}{2}|H(j\omega)|^2 = \frac{a}{2(1+\omega^2)}.$$

This is maximized for $\omega = 0$, so the gain of entrainment is at least $c_T(1) = a/2$. Observe that examples with arbitrarily large gain of entrainment can be obtained by taking the constant $a$ in (2.8) large enough.

In the next section, we cast the problem of determining the gain of entrainment as an optimal control problem.

# 3. Optimal control formulation

Consider the control system (2.1) with $n$ state variables and $m$ inputs. We assume that the system entrains. Pick any $T > 0$ and any $q \in \mathbb{R}^m$. We restrict attention to $T$-periodic controls satisfying the individual weighted average constraints

$$\frac{1}{T}\int_0^T \Xi(t, u(t))\,\mathrm{d}t = q, \tag{3.1}$$

where $\Xi:[0, T] \times \mathbb{R}_+^m \to \mathbb{R}_+^m$ is an integrable vector positive function that satisfies $\frac{1}{T}\int_0^T \Xi(t, q)\,\mathrm{d}t = q$.

The $n$-dimensional control system with the integral constraint on the controls (3.1) can be lifted to an $(n + m)$-dimensional nonlinear control system by adding $m$-equations to (2.1):

$$\dot{\tilde{x}} = \begin{bmatrix} \dot{x} \\ \dot{\xi} \end{bmatrix} = \begin{bmatrix} f(x, u) \\ \Xi(t, u) \end{bmatrix} = F(t, \tilde{x}, u), \tag{3.2}$$

where $\tilde{x} := [x^T\ \xi^T]^T$. We impose the boundary conditions

$$x(0) = x(T), \quad \xi(0) = 0, \ \xi(T) = Tq. \tag{3.3}$$

Since we consider systems that entrain, for any $T$-periodic control, there corresponds a unique GAS $T$-periodic solution $\gamma_T(t)$. The condition $x(0) = x(T)$ guarantees that the maximization is performed over this solution. The other two conditions are equivalent to (3.1).

To make the problem well posed, we will assume that controls take values in a hypercube $[\ell, L]^m$, where $0 < \ell < L$. We then formulate an optimal control problem as follows:

**Problem 3.1.** Fix values $0 < \ell < q < L$. Find an admissible control $u$ that maximizes the objective functional

$$J(u) := \frac{1}{T}\int_0^T h(t, x(t), u(t))\,\mathrm{d}t,$$

subject to the ODE (3.2), the constraint (3.1), $u_j(t) \in [\ell, L]$, $j = 1, \ldots, m$, $t \in [0, T]$, and the boundary conditions (3.3).

Note that $J(u)$ is the average value of the output along the globally attractive $T$-periodic solution (recall that we assume that $h$ is $T$-periodic in its first variable).

In what follows, we always consider systems affine in the control. Then, the fact that $[\ell, L]^m$ is compact and convex implies, by Filippov's theorem (e.g. [49]), that the reachable set at any time $t \geq 0$ is compact. Since $h$ is locally Lipschitz, an optimal control exists.

The optimal control formulation allows to apply powerful theoretical tools for solving optimal control problems as well as use software packages for numerical solutions (e.g. [50,51]). In the next section, we demonstrate how to determine the gain of entrainment using this formulation for both time-invariant and time-varying cost functions.

**Remark 3.2.** The control signals can be assumed to belong to different intervals. In other words, we can have $u_j(t) \in [\ell_j, L_j], j = 1, \ldots, m$. Nevertheless, we have simplified the aforementioned formulation by re-scaling the controls, so that they all satisfy the same bounds.

The following result is immediate:

**Theorem 3.3.** *A control u is a solution of problem 3.1 iff it maximizes $c_T(u) = z_T(u) - q$ as defined in (2.4).*

In other words, the problem of choosing the best periodic input signal has been transformed into a standard optimal control problem. Hence, to find how much improvement we get with periodic inputs, i.e. the gain of entrainment, we must find a control $u$ that solves problem 3.1 and then compute $J(u) - q$.

## 3.1. Pontryagin's maximum principle

Problem 3.1 can be studied in the framework of the PMP [40–42]. The *Hamiltonian* associated with our problem is

$$\mathcal{H}(t, u, \tilde{x}, p, p_0) := p^T(t)F(t, \tilde{x}, u) + \frac{p_0}{T}h(t, x, u),\tag{3.4}$$

where $p(t) \in \mathbb{R}^{n+m}$ is the co-state, and the *abnormal multiplier* $p_0 \geq 0$ is a constant.

**Proposition 3.4 (PMP).** *Let $u^*(t) \in \mathbb{R}^m, t \geq 0$ be an optimal control for problem 3.1, and let $\tilde{x}^* : [0, T] \to \mathbb{R}^{n+m}$ be the corresponding optimal trajectory. There exist $p_0^* \geq 0$ and $p^* : [0, T] \to \mathbb{R}^{n+m} \setminus \{0\}$, such that:*

1. *The optimal state $\tilde{x}^*(t)$ and corresponding adjoint $p^*(t)$ satisfy:*

$$\left.\begin{array}{l}\dot{\tilde{x}}^* = \dfrac{\partial \mathcal{H}}{\partial p}(t, u^*, \tilde{x}^*, p^*, p_0^*) \\[2mm] \dot{p}^* = -\dfrac{\partial \mathcal{H}}{\partial \tilde{x}}(t, u^*, \tilde{x}^*, p^*, p_0^*).\end{array}\right\}\tag{3.5}$$

and

2. *The control $u^*(t)$ satisfies*

$$\mathcal{H}(t, s, \tilde{x}^*(t), p^*(t), p_0^*) \leq \mathcal{H}(t, u^*(t), \tilde{x}^*(t), p^*(t), p_0^*)\tag{3.6}$$

*for all $s \in [\ell, L]^m$ and almost every (a.e.) $t \in [0, T]$ .*
3. *The adjoint satisfies the transversality condition*

$$p_i^*(0) = p_i^*(T), \quad i = 1, \ldots, n.\tag{3.7}$$

4. $H(t, u^*(t), \tilde{x}^*(t), p^*(t), p_0^*) = 0$ *for all $t \in [0, T]$.*

The proof is given in appendix A. Use of the PMP to deduce the structure of the optimal control is a difficult problem in general. We will show in the next section how it can be used in certain cases.

A trajectory $\mathscr{X} := (u(t), \tilde{x}(t), p(t))$ is said to be *feasible* if it satisfies the ODEs (3.5) and the boundary conditions (3.3) and (3.7). A feasible trajectory $\mathscr{X}$ is an *extremal trajectory* if it satisfies the PMP, i.e. if it also satisfies proposition 3.4. Observe that any optimal trajectory must be an extremal by proposition 3.4.

# 4. Occupancy models with controlled inflow and outflow

In this section, we look more closely at the $n$-dimensional occupancy model of the form

$$\begin{bmatrix} \dot{x}_1 \\ \dot{x}_2 \\ \vdots \\ \dot{x}_{n-1} \\ \dot{x}_n \end{bmatrix} = \begin{bmatrix} u_0(t)(1-x_1) - \lambda_1 x_1(1-x_2) \\ \lambda_1 x_1(1-x_2) - \lambda_2 x_2(1-x_3) \\ \vdots \\ \lambda_{n-2} x_{n-2}(1-x_{n-1}) - \lambda_{n-1} x_{n-1}(1-x_n) \\ \lambda_{n-1} x_{n-1}(1-x_n) - u_1(t)x_n \end{bmatrix}. \tag{4.1}$$

This is an $n$-dimensional RFM with initiation and exit rates that are non-negative control inputs. Suppose that both $u_0(t)$, $u_1(t)$ are periodic with period $T \ge 0$. It was proved in [6] that the RFM with $T$-periodic rates entrains, so in particular, (4.1) admits a unique solution $\gamma_T(t)$, with $\gamma_T(0) = \gamma_T(T)$, and $x \to \gamma_T$ for any initial condition $x(0) \in [0, 1]^n$.

To study the cost of entrainment in this system, fix $0 < \ell < L$. We will assume that the rates $u_0(t)$, $u_1(t)$ are two measurable and essentially locally bounded functions taking values in the interval $[\ell, L]$.

To allow a 'fair' comparison between $T$-periodic controls and constant controls, fix two values $\bar{u}_0, \bar{u}_1 \in (\ell, L)$ and pose integral constraints on the controls

$$\frac{1}{T} \int_0^T \alpha_0(t) u_0(t) \, dt = \bar{u}_0 \quad \text{and} \quad \frac{1}{T} \int_0^T \alpha_1(t) u_1(t) \, dt = \bar{u}_1, \tag{4.2}$$

for some given positive measurable functions $\alpha_0(t)$, $\alpha_1(t)$ satisfying

$$\frac{1}{T} \int_0^T \alpha_0(t) \, dt = \frac{1}{T} \int_0^T \alpha_1(t) \, dt = 1.$$

We now use theorem 3.3 to formulate an optimal control problem that allows finding the gain of entrainment. Introduce the two-dimensional state $\xi := [x_{n+1} \, x_{n+2}]^T$ and let $u := [u_0 \, u_1]^T$. Then, the extended system is an $(n+2)$-dimensional nonlinear control system

$$\dot{x} := \begin{bmatrix} \dot{x}_1 \\ \dot{x}_2 \\ \vdots \\ \dot{x}_n \\ \dot{x}_{n+1} \\ \dot{x}_{n+2} \end{bmatrix} = f(x) + g(x)u(t), \tag{4.3}$$

where

$$f(x) := \begin{bmatrix} -\lambda_1 x_1(1-x_2) \\ \lambda_1 x_1(1-x_2) - \lambda_2 x_2(1-x_3) \\ \vdots \\ \lambda_{n-1} x_{n-1}(1-x_n) \\ 0 \\ 0 \end{bmatrix} \quad \text{and} \quad g(x) := \begin{bmatrix} 1-x_1 & 0 \\ 0 & 0 \\ \vdots & \\ 0 & -x_n \\ \alpha_0(t) & 0 \\ 0 & \alpha_1(t) \end{bmatrix}, \tag{4.4}$$

and the boundary conditions are as follows:

$$x_i(T) = x_i(0), \; i = 1, \ldots, n, \; x_{n+1}(0) = 0, \; x_{n+1}(T) = T\bar{u}_0, \; x_{n+2}(0) = 0, \; x_{n+2}(T) = T\bar{u}_1. \tag{4.5}$$

The following is the optimal control problem.

**Problem 4.1.** Find $u_0, u_1 : [0, T] \to [\ell, L]$ that maximize the cost functional

$$J(u_0, u_1) := \frac{1}{T} \int_0^T \beta(t) u_1(t) x_n(t) \, dt, \tag{4.6}$$

subject to the ODE (4.3), integral constraints (4.2) and the boundary conditions (4.5), where $\beta : [0, T] \to \mathbb{R}$ is a given non-negative measurable function.

In general, this seems to be a non-trivial problem. Nevertheless, this formulation allows the use of both theoretical and numerical optimal control tools, and we will provide exact results in special cases.

## 4.1. Application of the Pontryagin maximum principle

As in the previous section, we can write the Hamiltonian (3.4). In this case,

$$\mathcal{H}(u, x, p, p_0) := p^T(t)(f(x(t)) + g(x(t))u(t)) + \frac{p_0}{T}\beta(t)u_1(t)x_n(t), \tag{4.7}$$

which can be written as follows:

$$\mathcal{H} = \varphi_0(t)u_0(t) + \varphi_1(t)u_1(t), \tag{4.8}$$

where

$$\left.\begin{array}{l} \varphi_0(t) := p_1(t)(1 - x_1(t)) + \alpha_0(t)p_{n+1}(t) \\[2mm] \varphi_1(t) := x_n(t)\left(\dfrac{p_0}{T}\beta(t) - p_n(t)\right) + \alpha_1(t)p_{n+2}(t), \end{array}\right\} \tag{4.9}$$

and

are called the *switching functions*. These functions play a central role in the determination of the structure of the extremal solutions of the optimal control problem. In particular, the analysis based on the PMP is separated to two cases: first, the case when the switching functions are non-zero. In this case, each subtrajectory of an extremal trajectory is called a *regular arc*. Second, the switching functions are 'identically' zero in a sense that will be precisely defined. In such case, a subtrajectory of an extremal trajectory is called a *singular arc*.

If the Hamiltonian is linear in the control inputs, as in (4.7), then the regular arcs take a very simple form. The control inputs assume either the minimum or maximum values that are allowed for them. Such controllers are known as *bang-bang* controllers. This is a well-known result in optimal control. We state it for the sake of completeness, and the proof is provided in the appendix A.

Before stating the result, we need to define regular arcs rigorously. Let $\mathscr{X}$ be a feasible trajectory. Define the open set:

$$E_r := \{t \in [0, T] \mid \varphi_0(t)\varphi_1(t) \neq 0\}.$$

A *regular arc* is a restriction $\mathscr{X}|_V$ for some open subset $V \subset E_r$. The next result analyses regular arcs.

**Lemma 4.2.** *Let $\mathscr{X}$ be an extremal trajectory. Then for any $t \in E_r$ and $i \in \{0, 1\}$, we have*

$$u_i^*(t) = \begin{cases} L, & \text{if } \varphi_i(t) > 0, \\ \ell, & \text{if } \varphi_i(t) < 0. \end{cases}$$

This means that at any time $t$ where $\varphi_i(t) \neq 0$, the corresponding $u_i^*(t)$ is a bang-bang control, i.e. it takes an extremal value.

Therefore, unless either of the switching functions vanish on a non-zero measure set, the optimal control is bang-bang, meaning that it has values in $\{\ell, L\}^2$ for almost all $t$.

The next section considers the unweighted problem 4.1 and shows that a singular arc satisfies the PMP on $[0, T]$. Furthermore, for the one-dimensional problem, any extremal trajectory cannot contain any regular arcs. In other words, it must be fully singular.

## 4.2. The unweighted optimal control problem

In this section, we consider the unweighted version of problem 4.1, that is, the case where $\alpha_0(t) = \alpha_1(t) = \beta(t) \equiv 1$ for all $t \in [0, T]$.

### 4.2.1. Constant controls satisfy the Pontryagin maximum principle

We first show that constant controls satisfy the necessary conditions for optimality. Note that this does *not* guarantee that such solutions are optimal.

**Theorem 4.3.** *Consider the unweighted problem 4.1. The constant controls $u_0(t) \equiv \bar{u}_0$, $u_1(t) \equiv \bar{u}_1$ satisfy proposition 3.4 (the PMP) with the corresponding switching functions identically zero, i.e. a fully singular trajectory is extremal.*

*Proof.* Let $z := [p_1 \dots p_n]^T$, i.e. the first $n$ entries of the adjoint state. Equation (3.5) yields

$$\dot{z} = -J^T(x, u)z - bu_1, \tag{4.10}$$

where $J$ is the Jacobian of the RFM (4.1) with respect to $x$, and $b := [0 \dots 0 \; p_0/T]^T$. Also, $\dot{p}_{n+1}(t) = \dot{p}_{n+2}(t) \equiv 0$.

It has been shown in [29] that the RFM with constant rates admits a unique GAS steady state in $(0, 1)^n$. Hence, every solution of (4.1) with $u_0(t) \equiv \bar{u}_0$, $u_1(t) \equiv \bar{u}_1$, converges to a point $\bar{x} = [\bar{x}_1 \; \bar{x}_2 \; \dots \; \bar{x}_n]^T \in (0, 1)^n$. It was shown in [6] that if $M$ is any compact subset of $(0, 1)^n$, then there exists a matrix measure $\mu : \mathbb{R}^{n \times n} \to \mathbb{R}$ such that $\mu(J(x, u)) < 0$ for all $x \in M$, $u \geq 0$. This implies in particular that all the eigenvalues of $J(x, u)$ have a negative real part [52], so $J(x, u)$ is non-singular for each $x \in (0, 1)^n$, $u \geq 0$. Hence, so is $J^T(\bar{x}, \bar{u})$. Let $\bar{z} := -(J^T(\bar{x}, \bar{u}))^{-1} b \bar{u}_1$, and let $\bar{u} := [\bar{u}_0 \; \bar{u}_1]^T$. We now show that for

$$u(t) \equiv \bar{u}, \; x(t) \equiv \bar{x}, \; p_0 = T, \; p(t) \equiv \left[ \bar{z} \quad -\bar{p}_1(1 - \bar{x}_1)/\bar{u}_0 \quad -\bar{x}_n(1 - \bar{p}_n)/\bar{u}_1 \right]^T,$$

all the conditions in the PMP hold. First note that the boundary conditions (4.5) all hold. Equation (4.10) holds by the definition of $\bar{p}$. The switching functions (4.9) satisfy $\varphi_0(t) \equiv \varphi_1(t) \equiv 0$. Equation (4.8) implies that $\mathcal{H} \equiv 0$ and that (3.6) trivially holds. ∎

We have shown that constant controls satisfy the necessary conditions of the PMP. In other words, constant controls are always extremal solutions.

In the following section, we show that for $n = 1$, constant controls are the *only* controls that satisfy the PMP.

### 4.2.2. Extremal analysis of the one-dimensional unweighted problem

In this section, we study the following system:

$$\left. \begin{array}{l} \dot{x}_1(t) = u_0(t)(1 - x_1(t)) - u_1(t)x_1(t), \\ \dot{x}_2(t) = u_0(t) \\ \dot{x}_3(t) = u_1(t). \end{array} \right\} \tag{4.11}$$

and

The controls $u_0$ [$u_1$] represent time-varying initiation [exit] rates in an RFM with $n = 1$. Even though the PMP provides a general approach for addressing optimal control problems, it seldom leads to a full characterization of extremal solutions, especially in the case of multiple inputs. We will show that this is possible for the system (4.11): a detailed analysis using the PMP shows that any extremal trajectory corresponds to a constant $x_1(t)$. Since each control input takes values in a compact and convex set, the optimal control problem always has a solution. Thus, there is no gain of entrainment. This shows that the PMP is a viable approach for handling such problems and lays the ground for future generalization to higher dimensional cases.

**Theorem 4.4.** *Let $\mathscr{X}$ be an extremal trajectory for problem 4.1 with the system (4.11). Then*

$$x_1^*(t) \equiv \frac{\bar{u}_0}{\bar{u}_0 + \bar{u}_1} = \frac{1}{1 + (\bar{u}_1/\bar{u}_0)} \text{ for all } t \in [0, T]. \tag{4.12}$$

The proof is given in appendix A.

The PMP immediately yields the following result, which implies that periodic inputs do not confer any advantage over constant counterparts, and hence, the presence of periodic signals in an occupancy system cannot be a result of optimizing the unweighted throughput of the system.

**Theorem 4.5.** *Fix $0 < \ell < L$, and let the admissible controls $u_0$, $u_1$ take values in $[\ell, L]$, with given averages $\bar{u}_0$, $\bar{u}_1 \in (\ell, L)$, the optimal objective for problem 4.1 and system (4.11) is*

$$J^* = \frac{\bar{u}_0}{\bar{u}_0 + \bar{u}_1}. \tag{4.13}$$

*The optimal trajectory is $x_1^*(t) \equiv \bar{u}_0/(\bar{u}_0 + \bar{u}_1)$, a constant, and it can be achieved by the constant inputs:*

$$u_0^*(t) \equiv \bar{u}_0, \; u_1^*(t) \equiv \bar{u}_1. \tag{4.14}$$

**Remark 4.6.** The control inputs $u_0(t)$, $u_1(t)$ that achieve the optimal cost are not unique. Indeed, it is clear that the optimal solution in (4.12) depends only on the ratio $\bar{u}_1/\bar{u}_0$. For instance, $u_0^{**}(t) \equiv \bar{u}_0 \rho(t)$, $u_1^{**}(t) \equiv \bar{u}_1 \rho(t)$ is also an optimal solution for any function $\rho$ such that $\bar{u}_0 \rho(t) \subset [\ell, L]$ and $\bar{u}_1 \rho(t) \subset [\ell, L]$ for all $t \in [0, T]$.

**Remark 4.7.** Theorem 4.5 can be also proven via a more direct approach, motivated by an idea from [53]. This alternative proof is given in appendix A.

**Remark 4.8.** In the more specialized case of a single controller $u_0(t)$, that is, with $u_1$ fixed, techniques similar to the ones used in [46] can be used to generalize theorem 4.5 to scalar systems of the form $\dot{x} = u_0(t)g(x) - f(x)$ for any two $C^1$ functions $f, g : [0, 1] \to \mathbb{R}_{\geq 0}$ that satisfy: $f(0) = 0$, $g(1) = 0$, $f$, $g$ positive over $(0, 1)$, and $f(x)/g(x)$ is convex.

### 4.2.3. The unweighted problem for the ribosome flow model with $n = 2$ and a single input

We now study problem 4.1 for an RFM with dimension $n = 2$ and a single control $u_0(t)$ as the initiation rate, i.e.

$$\left.\begin{aligned}
\dot{x}_1 &= u_0(1 - x_1) - \lambda_1 x_1(1 - x_2), \\
\dot{x}_2 &= \lambda_1(1 - x_2)x_1 - \lambda_2 x_2 \\
\dot{x}_3 &= u_0(t),
\end{aligned}\right\} \tag{4.15}$$

and

subject to the boundary conditions (4.5). Here, our goal is to maximize the average value of the output rate $\lambda_2 x_2$.

More formally, we aim at solving the following problem:

**Problem 4.9.** Let $\ell$, $\bar{u}_0$, $L$ be given such that $0 < \ell < \bar{u}_0 < L$. Find $u_0 : [0, T] \to [\ell, L]$ that maximizes the cost functional $J(u_0) := \frac{1}{T} \int_0^T \lambda_2 x_2(t) \, dt$, that is, the average production rate subject to the ODE (4.15), integral constraint $\frac{1}{T} \int_0^T u_0(t) = \bar{u}_0$, and the boundary conditions $x_1(0) = x_1(T)$, $x_2(0) = x_2(T)$.

The next result shows that here as well a constant control is optimal, that is, there is no gain of entrainment.

**Theorem 4.10.** Fix $0 < \ell < L$ and $\bar{u}_0 \in (\ell, L)$, the objective function for problem 4.9 with the system (4.15) is maximized by the constant control $u_0^*(t) \equiv \bar{u}_0$.

*Proof.* The first equation in (4.15) is the first equation (4.11) for $u_1(t) = \lambda_1(1 - x_2)$, so $\bar{u}_1 = \lambda_1(1 - \bar{x}_2)$, and the output rate is $\lambda_1(1 - x_2)x_1$. (Note that here $x_2$ is the second state-variable in (4.15), and not the integral of $u_0$ as in (4.11)). By theorem 4.5, we have $\lambda_1 \overline{u_1 x_1} \leq (\lambda_1 \bar{u}_1 \bar{u}_0)/(\bar{u}_0 + \lambda_1 \bar{u}_1)$. Hence,

$$\lambda_1 \overline{u_1 x_1} = \lambda_1 \overline{(1 - x_2)x_1} \leq \frac{\lambda_1 \bar{u}_1 \bar{u}_0}{\bar{u}_0 + \lambda_1 \bar{u}_1} = \frac{\lambda_1(1 - \bar{x}_2)\bar{u}_0}{\bar{u}_0 + \lambda_1(1 - \bar{x}_2)}. \tag{4.16}$$

Integrating (4.15), we get

$$0 = x_2(T) - x_2(0) = \int_0^T \dot{x}_2(t) \, dt = \lambda_1 \int_0^T (1 - x_2(t))x_1(t) \, dt - \int_0^T \lambda_2 x_2(t) \, dt.$$

Hence, $\lambda_1 \overline{(1 - x_2)x_1} = \lambda_2 \bar{x}_2$. Substituting in (4.16), we get

$$\lambda_2 \bar{x}_2 \leq \frac{\lambda_1 \bar{u}_0(1 - \bar{x}_2)}{\lambda_1(1 - \bar{x}_2) + \bar{u}_0}. \tag{4.17}$$

The left-hand side here is the quantity that we are trying to maximize. Rearranging gives:

$$f(\bar{x}_2) \geq 0, \tag{4.18}$$

where $f(s) := s^2 - (1 + \bar{u}_0(1/\lambda_1 + (1/\lambda_2)))s + (\bar{u}_0/\lambda_2)$. Let $p$, $q$ denote the roots of $f(s)$. Then, $f(s) = (s - p)(s - q)$ gives

$$\left.\begin{aligned}
1 + \bar{u}_0\left(\frac{1}{\lambda_1} + \frac{1}{\lambda_2}\right) &= p + q \\
\frac{\bar{u}_0}{\lambda_2} &= pq.
\end{aligned}\right\} \tag{4.19}$$

and

Recall that the RFM (4.15) with the constant control $u_0(t) \equiv \bar{u}_0$ admits a *unique* steady state $\bar{e} \in (0, 1)^2$ [29]. It is straightforward to show that $f(\bar{e}_2) = 0$. We may assume that $p = \bar{e}_2$, so $p \in (0, 1)$. Then, (4.19) implies that $q$ is real and $q > 1$. The quadratic inequality (4.18) implies that either $\bar{x}_2 \leq p < 1$ or $\bar{x}_2 \geq q > 1$. Since $x_2(t) \in [0, 1]$ for all $t$, the second inequality can be ignored, and we have that the maximal (feasible) value of $\bar{x}_2$ is $\bar{x}_2^* = p = e_2$. Obviously, this is attained for the constant control $u_0(t) \equiv \bar{u}_0$. ∎

## 4.3. Gain of entrainment with time-varying weight functions

Analysing the case with time-varying weights is challenging, but it is highly relevant to applications since resources may be allocated differently during the period. In this section, we show that, even for the above examples, once the weighting functions become time varying, constant inputs may no longer be optimal.

We consider the special case of (4.3) with $n = 1$ and a single input $u_0(t)$ as the initiation rate, i.e.

$$\begin{aligned} \dot{x}_1(t) &= u_0(t)(1 - x_1(t)) - \lambda_1 x_1(t) \\ \dot{x}_2(t) &= u_0(t). \end{aligned}\right\}$$
and
(4.20)

Our goal now is to maximize $J(u_0) := (\lambda_1/T) \int_0^T \beta(t) x_1(t) \, dt$, subject to the boundary conditions (4.5), and where the weight function $\beta$ is differentiable and satisfies $\beta(t) > 0$ for all $t \in [0, T]$. Without loss of generality, we assume that $T = 1$.

**Proposition 4.11.** *Suppose that $u_0^*$ is an optimal control. Then, for almost all $t \in [0, 1]$ we have that either $u_0^*(t) \in \{\ell, L\}$ or*

$$u_0^*(t) = c\sqrt{\beta(t)} - \lambda_1 + \frac{\dot{\beta}(t)}{2\beta(t)}, \tag{4.21}$$

*for some constant $c$. Furthermore, if $u_0^*$ satisfies (4.21) for all $t \in [0, 1]$, then*

$$c = \frac{\bar{u}_0 + \lambda_1 - \frac{1}{2}\log(\beta(1)/\beta(0))}{\int_0^1 \sqrt{\beta(t)} \, dt}. \tag{4.22}$$

**Remark 4.12.** Note that if $\bar{u}_0 \in (\ell, L)$, then this implies that the constant control $u_0(t) \equiv \bar{u}_0$ cannot be optimal. If (4.21) does not hold for any $t$ (e.g. when the right-hand side of (4.21) takes values that are not in $[\ell, L]$), then the optimal control is bang-bang.

As a specific example take

$$\ell = 0.001, L = 10, \ \bar{u}_0 = 2, \ \lambda_1 = 1,$$

and the weight function

$$\beta(t) = e^{-\rho(t - (T/2))^2},$$

with $\rho = 100$ and $T = 1$. In the context of the RFM, this would represent the case where it is required to highly express a specific protein near the middle of every cycle, rather than having a uniform level of production along the entire period.

The constant input $u_0(t) \equiv 2$ yields a steady-state trajectory $x(t) \equiv 2/3$. The corresponding value of the objective function is expressed as follows:

$$J(u_0) = \frac{2}{3} \int_0^1 \beta(t) \, dt = 0.118164. \tag{4.23}$$

Our optimal control formulation can be used to solve the problem *numerically* using optimal control packages like shown in [50]. The result is a three-arc bang-bang control

$$u^*(t) = \begin{cases} \ell, & t \in [0, t_1), \\ L, & t \in [t_1, t_1 + \Delta), \\ \ell, & t \in [t_1 + \Delta, 1], \end{cases}$$

where $\Delta := (\bar{u}_0 - \ell)/(L - \ell) = 0.19992$ and $t_1 = 0.27335$. The corresponding periodic solution satisfies $x(1) = x(0) = 0.50251$. This achieves a cost $J(u^*) = 0.141183$, which is roughly 20% better than (4.23).

Figure 3 depicts the approximate optimal bang-bang control (computed using [50] and a bisection procedure) and the resulting periodic solution $x_1^*(t)$. The weight function $\beta(t)$ is also shown. The maximal value of $\beta$ is achieved at $t = 1/2$. The optimal control switches to the maximal value $L = 10$ before the peak time of $\beta$. This makes sense as it guarantees that $x_1^*(t)$ will have large values when the weighting of $x_1^*(t)$ in the cost function is large.

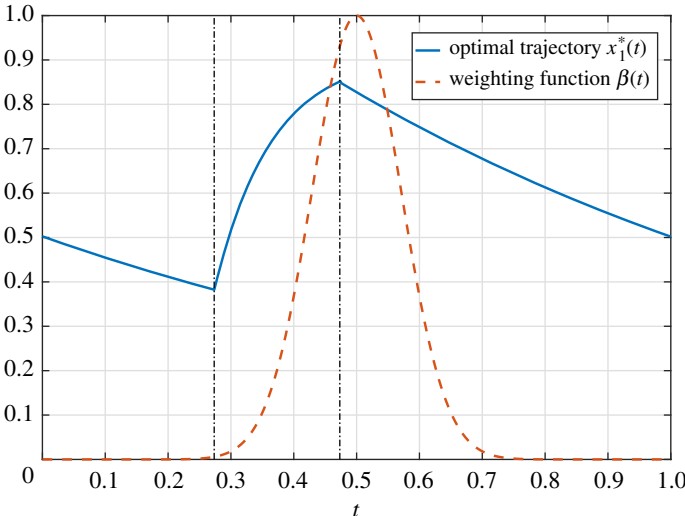

**Figure 3.** The optimal trajectory for maximizing a weighted throughput is *non-constant*. The plot shows the weighting function $\beta(t)$ with the corresponding optimal periodic solution $x_1^*(t)$. The vertical dashed lines denote the switching points of the three-arc bang-bang controller. Note that the high bang control is switched on when the value of the weight function becomes non-negligible.

# 5. Gain of entrainment in generalized occupancy models

In this section, we consider general cascades like those in figure 1. Recall that under the conditions (1.4), the $n$-dimensional RFM can be approximated by (1.5), which has the form in figure 1$a$. We consider here this approximated system.

## 5.1. Unweighted objective functional

We first state the following result which is well known in the theory of linear time-invariant systems (see also [24]). We include the proof for completeness.

**Proposition 5.1.** *Consider a single-input-single-output linear system:* $\dot{z} = Az + bw$, $y = c^T z$, *where* $z \in \mathbb{R}^n$, *and A is Hurwitz. Let w be a bounded measurable input which is T-periodic. Then, y converges to a steady-state T-periodic solution* $y_w$, *and*

$$\int_0^T y_w(t) \, \mathrm{d}t = H(0) \int_0^T w(t) \, \mathrm{d}t,$$

*where* $H(s) := c^T(sI - A)^{-1}b$ *is the transfer function of the linear system.*

*Proof.* Since $w$ is measurable and bounded, $w \in L_2([0, T])$. Hence, it can be written as a Fourier series $w(t) = \bar{w} + \sum_i a_i \sin(\omega_i t + \phi_i)$. The output of the linear system converges to the steady-state periodic solution

$$y_w(t) = H(0)\bar{w} + \sum_i |H(j\omega_i)| a_i \sin(\omega_i t + \phi_i + \angle(H(j\omega_i))).$$

Each sinusoid in the expansion has period $T$, so $\int_0^T y_w(t) \, \mathrm{d}t = T H(0)\bar{w}$. ∎

Combining proposition 5.1 with our results on the gain of entrainment in certain bottleneck models yields the following result.

**Theorem 5.2.** *Consider the nonlinear system depicted in either* figure 1$a$ *or* figure 1$b$ *with A Metzler, and b,* $c \in \mathbb{R}_+^n$. *Let* $u_0(t)$, $u_1(t)$ *be T-periodic non-negative control signals. For any* $0 < \ell < L$ *and any* $\bar{u}_0, \bar{u}_1 \in (\ell, L)$, *consider the functional*

$$J(u_0, u_1) := \frac{1}{T} \int_0^T w_2(t) \, \mathrm{d}t,$$

**Figure 4.** A generalized occupancy model with a two-dimensional bottleneck entrance. The controls are the scalar functions $u_0(t)$, $u_1(t)$. We have $x_1, x_2, x_3, w_1, w_2, y \in \mathbb{R}_+, z \in \mathbb{R}^n, A \in \mathbb{R}^{n \times n}, b, c \in \mathbb{R}^n_+$. The linear system block is assumed to be positive and Hurwitz.

where $w_2(t)$ is the steady-state $T$-periodic output signal. Then, the constant controls

$$u_0^*(t) \equiv \bar{u}_0 \quad \text{and} \quad u_1^*(t) \equiv \bar{u}_1. \tag{5.1}$$

maximize $J$.

*Proof.* Consider the system depicted in figure 1a. Fix admissible $T$-periodic controls $u_0(t)$, $u_1(t)$. Denote the corresponding steady-state average values of $w_1(t)$ and $y(t)$ by $\bar{w}_1$ and $\bar{y}$. Obviously, $\bar{w}_2 = \bar{y}$. By proposition 5.1, $\bar{w}_2 = H(0)\bar{w}_1$, where $H$ is the transfer function of the linear system. Since $A$ is Metzler and $b, c \in \mathbb{R}^n_+$, the trajectories of the linear system are positive. Thus, maximizing $J$ is equivalent to maximizing $\bar{w}_1$. Theorem 4.5 implies that the constant controls $u_0^*(t) \equiv \bar{u}_0$ and $u_1^*(t) \equiv \bar{u}_1$ maximize $\bar{w}_1$. The system in figure 1b can be treated similarly. ∎

**Remark 5.3.** The same result holds if the single-input modules in figure 1 are replaced by the single-input RFM with $n = 2$ in (4.15) as shown in figure 4.

## 5.2. Weighted objective functional

We extend the results of §4.3 to the GOM in figure 1a. We show that constant rates are no longer optimal. As before, let $\beta$ be a differentiable and positive weight function defined over $[0, T]$.

**Theorem 5.4.** *Consider the nonlinear system depicted in either* figure 1a *with A Metzler, and b, $c \in \mathbb{R}^n_+$. Let $u_0(t)$ be a $T$-periodic non-negative control signal, and let $u_1(t) = \beta(t)$ be positive and differentiable over $[0, T]$. For any $0 < \ell < L$ and any $\bar{u}_0 \in (\ell, L)$, consider the functional $J(u_0) := \frac{1}{T} \int_0^T w_2(t)\, dt$, where $w_2(t)$ is the steady-state $T$-periodic output signal. Suppose that $u_0^*$ is an optimal control. Then for almost all $t \in [0, 1]$, we have that either $u_0^*(t) \in \{\ell, L\}$ or it satisfies (4.21) for some constant c. Furthermore, if $u_0^*$ satisfies (4.21) for all $t \in [0, 1]$, then c can be computed by (4.22).*

*Proof.* The proof follows immediately from theorem 4.11 and proposition 5.1. ∎

# 6. Discussion

Entrainment is an interesting and important property of dynamical systems. It allows systems to develop an 'internal clock' that synchronizes to periodic excitations like the 24 h solar day. Such clocks are important in biology, as they allow organisms to adequately respond to periodic processes like the solar day and the cell cycle division process. They are also essential for synthetic biology, as a common clock is an important ingredient in building synthetic biology circuits that include several modules working in synchrony.

Here, we considered an additional qualitative property called the *gain of entrainment*. This measures the advantage, if any, of using a periodic control vs. an 'equivalent' constant control for maximizing the average throughput. We showed how this problem can be cast as an optimal control problem. This allows using the sophisticated analytical and numerical tools developed for solving optimal control problems to determine the gain of entrainment.

We have shown that, perhaps surprisingly, there is no gain of entrainment in a class of systems relevant to biology and traffic applications. In other words, in these systems, non-constant periodic controls are no better than constant controls. The optimality of constant controls fails to hold if we allow *time-varying* objective functionals. This result is not particularly surprising and has been recapitulated in other models [37], where it has been shown that that gain of entrainment is positive when the system exhibits two phases such as day and night. Hence, this suggests the possibility that the observation of non-constant periodic signals in biological systems is correlated with the

maximization of the throughput with varying weights along each period or cycle. The other possibility is that periodic signals are driven by processes external to the occupancy system. Distinguishing between these two scenarios is a subject of future research.

Future work includes identifying general classes of systems where the gain of entrainment is positive and trying to understand the structure of the optimal periodic controls. The literature on optimal fishing [38] has revealed that optimal solutions are periodic when the fishermen are allowed to target fish in a specific age group only. This may be interpreted as a kind of periodic constraint on the control. It may be of interest to understand if such constraints also appear in the context of cellular systems.

Another interesting research direction is to generalize these results to models such as the nonlinear $n$ site RFM with $(n + 1)$ time-periodic control inputs.

Data accessibility. This article has no additional data.

Authors' contributions. M.M. and E.D.S.: initiated the project and acquired the funding; M.A.A.: did the first draft of mathematical proofs and simulations; M.A.A. and M.M.: wrote the first draft. All the authors discussed the details and reviewed the manuscript.

Competing interests. We declare we have no competing interests.

Funding. This research has been supported in part by research grants from the Israel Science Foundation, the US-Israel Binational Science Foundation, ONR N00014-21-1-2431, AFOSR FA9550-21-1-0289, NSF 2052455 and 1817936.

Acknowledgements. The authors thank Mahdiar Sadeghi for helpful discussions.

# Appendix A. Additional Proofs

## A.1. Proof of proposition 3.4

Most of the statements here are the standard PMP. We only need to prove the transversality condition (3.7).

Pick a set $S \subseteq \mathbb{R}^{2(n+m)}$, and suppose that the state must satisfy the constraint $\left[ x(0)^T x(T)^T \right]^T \in S$. Then, the corresponding transversality condition [40] is given as follows:

$$\begin{bmatrix} p(0) \\ -p(T) \end{bmatrix} \perp \mathcal{T}_{\begin{bmatrix} x(0) \\ x(T) \end{bmatrix}} S,$$

where $\mathcal{T}_z S$ denotes the tangent space of $S$ at $z$. In our case, (3.3) gives

$$S = \{z \in \mathbb{R}^{2n+2m} \mid [z_1, \ldots, z_n]^T = [z_{n+m+1}, \ldots, z_{2n+m}]^T,$$

$$[z_{n+1}, \ldots, z_{n+2}]^T = 0, [z_{2n+m+1}, \ldots, z_{2n+2m}]^T = Tq\}.$$

Hence, $T_z S = \text{span}\{v^1, \ldots, v^n\}$, where $v^i$ is the vector with one at entries $i$ and $(i+n+m)$, and zero elsewhere. Therefore, it is necessary that $p_i^*(0) = p_i^*(T)$, $i = 1, \ldots, n$.

## A.2. Proof of lemma 4.2

We prove the result for $i = 0$. (The proof for $i = 1$ is very similar.) Suppose that $\varphi_0(t) > 0$ for some $t \in [0, T]$. Seeking a contradiction, suppose that $u_0^*(t) < L$. Then,

$$\mathcal{H}(u_0^*(t), u_1^*(t), x^*(t), p^*(t)) = \varphi_1(t)u_1^*(t) + \varphi_0(t)u_0^*(t)$$
$$< \varphi_1(t)u_1^*(t) + \varphi_0(t)L$$
$$= \mathcal{H}(L, u_1^*(t), x^*(t), p^*(t)),$$

and this contradicts (3.6). Hence, $u_0^*$ is not optimal. The same argument can be applied when $\varphi_0(t) < 0$.

## A.3. Proof of theorem 4.4

The proof, based on the analysis of extremals, is divided into a sequence of lemmas. For a set $A \subset \mathbb{R}$, $\mu(A)$ denotes its Lebesgue measure. The set of accumulation points of $A$ is denoted by $A'$. For $x \in \mathbb{R}$, $\{x\} + A := \{x + a \mid a \in A\}$.

Recall that we consider (4.11) , so

$$\mathcal{H} = p_1(u_0(1-x_1) - u_1 x_1) + p_2 u_0 + p_3 u_1 + \frac{p_0}{T} u_1 x_1.$$

**Lemma A.1.** *The adjoint variables $p_i^*$ satisfy*:

$$\left.\begin{array}{c} \dot{p}_1^*(t) = -(u_0(t) + u_1(t))p_1^*(t) - u_1(t), \\ p_2^*(t) \equiv p_2^*(0) \\ p_3^*(t) \equiv p_3^*(0), \end{array}\right\} \tag{A 1}$$

and

*with the boundary condition $p_1^*(0) = p_1^*(T)$.*

*Proof.* We first show that we can take $p_0^* = T$. Assume that $p_0^* = 0$. Then (3.5) yields $\dot{p}_1^* = (u_0^*(t) + u_1^*(t))p_1^*$. Integrating over $[0, T]$, we get

$$\log|p_1^*(T)| - \log|p_1^*(0)| = \int_0^T (u_0^*(t) + u_1^*(t)) \, dt = T(\bar{u}_0 + \bar{u}_1).$$

By the transversality condition, we know that $p_1^*(0) = p_1^*(T)$, which implies that $\bar{u}_0 + \bar{u}_1 = 0$. This is a contradiction, so we conclude that $p_0^* > 0$, and by scaling the objective function we may take $p_0^* = T$. Now (A1) follows from calculating the partial derivatives in (3.5). ∎

## A.3.1. Analysis of the switching functions

Using lemma A.1, the switching functions in our case are as follows:

$$\varphi_0(t) = p_1(t)(1 - x_1(t)) + p_2(0) \tag{A 2}$$

and

$$\varphi_1(t) = x_1(t)(1 - p_1(t)) + p_3(0). \tag{A 3}$$

Given an extremal trajectory $\mathscr{X}$, let

$$E_+^i := \{t \in [0, T] \mid \varphi_i(t) > 0\},$$
$$E_-^i := \{t \in [0, T] \mid \varphi_i(t) < 0\},$$
$$E_0^i := \{t \in [0, T] \mid \varphi_i(t) = 0\},$$

where $i = 0, 1$. Note that $E_+^i, E_-^i, i = 0, 1$, are open relative to $[0, T]$, and $E_0^i, i = 0, 1$, are closed. In particular, all these sets are Lebesgue measurable.

A calculation gives

$$\dot{\varphi}_0(t) = u_1(t)(p_1(t) - (1 - x_1(t))) \tag{A 4}$$

and

$$\dot{\varphi}_1(t) = u_0(t)(1 - x_1(t) - p_1(t)). \tag{A 5}$$

**Remark A.2.** The functions $\varphi_0$, $\varphi_1$ are absolutely continuous. Hence, they are differentiable almost everywhere and have bounded derivatives. This implies that both $\varphi_0$, $\varphi_1$ are Lipschitz continuous. Also, since the controls are positive, (A 4) and (A 5) imply that $\mathrm{sgn}(\dot{\varphi}_0(t)) = -\mathrm{sgn}(\dot{\varphi}_1(t))$ whenever both $\varphi_0$ and $\varphi_1$ are differentiable.

## A.3.2. Characterization of singular arcs

In this section, we are interested in the case where $\mu(E_0^i) > 0$ for either $i = 0$ or $i = 1$. Let

$$E_s := \{t \in [0, T] \mid \varphi_0(t)\varphi_1(t) = 0\} = E_0^0 \cup E_0^1.$$

Let $\mathscr{X}$ be an extremal trajectory. We call any restriction of $\mathscr{X}$ to any non-zero-measure subset of $E_s$ a *singular arc*.

The following lemmas characterize the behaviour on singular arcs.

**Lemma A.3.** *Let $\mathscr{X}$ be an extremal trajectory, and assume that $\mu(E_0^i) > 0$ for some $i \in \{0, 1\}$. Then, there exists $c_i \in (0, 1)$ such that*

$$x_1^*(t) = c_i \text{ for almost all } t \in E_0^i.$$

*Furthermore, $\dot{x}_1^*(t) = 0$ for almost all $t \in E_0^i$ and the two inputs satisfy*

$$\left( \frac{1}{c_i} - 1 \right) u_0^*(t) = u_1^*(t), \quad \text{for almost all } t \in E_0^i. \tag{A 6}$$

*Proof.* Let $E_0^{i'} \subseteq E_0^i$ denote the set of accumulation points of $E_0^i$. Note that $\mu(E_0^{i'}) = \mu(E_0^i)$, since $E_0^i \setminus E_0^{i'}$ is the set of isolated points of $E_0^i$, which is countable and hence has measure zero. Let $F_i := \{t \in [0, T] \mid \dot{\varphi}_i^*(t) \text{ exists}\} \cap E_0^{i'}$. Then, $\mu(F_i) = \mu(E_0^i)$ since $\varphi_i^*$ is differentiable a.e.

Fix $t \in F_i$. By definition, we have $\varphi_i^*(t) = 0$. We show that $\dot{\varphi}_i^*(t) = 0$ as well. Since $t$ is an accumulation point, $\exists \{t_k\}_{k=1}^\infty \subset E_0^i$ such that $t_k \to t$. Since $\varphi_i^*$ is differentiable at $t$, $\dot{\varphi}_i^*(t) = \lim_{k \to \infty} (\varphi_i^*(t_k) - \varphi_i^*(t))/(t_k - t) = 0$. We consider the case $i = 0$ (the proof when $i = 1$ is very similar). Using (A 2) and (A 4), the equations $\varphi_0^*(t) = \dot{\varphi}_0^*(t) = 0$ yield $p_1^*(t)(1 - x_1^*(t)) = -p_2^*(0)$, and $p_1^*(t) = 1 - x_1^*(t)$. Since $x_1^*(t) \in (0, 1)$, $p_2^*(0) < 0$ and

$$x_1^*(t) = 1 - p_1^*(t) = c_0, \quad \text{for all } t \in F_0, \tag{A 7}$$

where $c_0 := 1 - \sqrt{-p_2^*(0)}$.

Let $F_0'$ be the set of accumulation points of $F_0$. Then, $\mu(F_0') = \mu(E_0^0)$. Fix $t \in F_0'$. Hence, $\exists \{t_k\}_{k=1}^\infty \subset F_i$ such that $t_k \to t$. Since $x_1^*(t_k) = c_0$ for all $k$, $\dot{x}_1^*(t) = 0$. Substituting this in (4.11) proves (A 6). ∎

The next result shows that if an extremal trajectory $\mathscr{X}$ has $x_1^*$ identically constant, then it satisfies (4.12), and $\mathscr{X}$ consists entirely of singular arcs.

**Lemma A.4.** *Let $\mathscr{X}$ be an extremal trajectory. If $x_1^*(t)$ is identically constant on $[0, T]$, then*

$$x_1^*(t) \equiv \frac{\bar{u}_0}{(\bar{u}_1 + \bar{u}_0)}. \tag{A 8}$$

*Furthermore, $\varphi_0^*(t)\varphi_1^*(t) \equiv 0$, i.e. $E_s = [0, T]$.*

*Proof.* By assumption, there exists $c \in (0, 1)$ such that $c \equiv x_1^*(t)$. Substituting this in (4.3) yields $\dot{x}_1^* = u_0^*(t) - (u_1^*(t) + u_0^*(t))c \equiv 0$, and integrating over $[0, T]$ proves (A 8). We also find that $u_1^*(t) \equiv (1c - 1)u_0^*(t)$, and substituting this in (A 1) gives $\dot{p}_1^*(t) = u_1^*(t)(-1 - 11 - cp_1^*(t))$. Combining this with the boundary condition (3.7) gives $p_1^*(t) \equiv c - 1$. Equations (A 2) and (A 3) give

$$\left. \begin{aligned} \phi_0(t) &= (c - 1)(1 - c) + p_2^*(0) \\ \phi_1(t) &= c(-c) + p_3^*(0), \end{aligned} \right\} \tag{A 9}$$

and

for all $t \in [0, T]$. To prove that $E_r = \emptyset$, we first assume that that $\mu(E_s) = 0$. Then, $E_r = [0, T]$ (a.e.), and equation (A 9) implies that both switching functions are constant and not zero, by the definition of $E_r$. Lemma 4.2 implies that every $u_i^*$ is constant, i.e. either $u_i^*(t) \equiv \ell$ or $u_i^*(t) \equiv L$. This contradicts the fact that $\bar{u}_i^* = \frac{1}{T} \int_0^T u_i^*(s)\, ds \in (\ell, L)$. Thus, $E_s = [0, T]$. We conclude that $\mu(E_s) > 0$. Pick $\tau \in E_s$. Then, $\phi_0(\tau) = \phi_1(\tau) = 0$ and (A 9) implies that $\phi_0(t) = \phi_1(t) = 0$ for all $t$, so $E_s = [0, T]$. ∎

## A.3.3. Inadmissibility of regular arcs

In the previous section, we decomposed an extremal trajectory into regular and singular arcs. On the regular arcs, the control is bang-bang. On the singular arcs, the controls satisfy (A 6) and the state must be constant almost everywhere, so $\dot{x}_1^*(t) = 0$ a.e. In general, an extremal trajectory can consist of an arbitrary patching of regular and singular arcs. In this section, we consider the admissibility of regular arcs.

To simplify presentation, we write $x$ and $p$ instead of $x_1$ and $p_1$ from here on. Furthermore, we denote extremal trajectories by $x^*$, $p^*$. Figure 5 depicts the dynamics of $x$ based on lemmas 4.2 and A.3.

**Lemma A.5.** *Let $\mathscr{X}$ be an extremal trajectory. Then, $x(t), p(t) \in (\ell/(L + \ell), L/(L + \ell))$ for all $t \in [0, T]$.*

*Proof.* If $x(t)$ is identically constant then the proof follows from lemma A.4. Hence, we assume that $x(t)$ is not constant. This implies that we can restrict attention to the four regular cases depicted in figure 5.

Suppose that $x(0) > L/(L + \ell)$. Then considering the regular cases depicted in figure 5, we see that $x(T) < L/(L + \ell)$, and this contradicts the periodicity condition $x(0) = x(T)$.

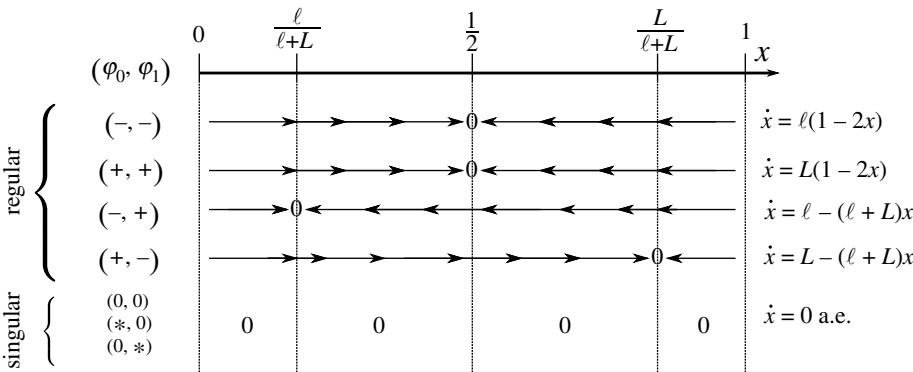

**Figure 5.** The equation for $\dot{x}$ and the directions of the dynamics as a function of $x \in (0, 1)$ for all possible arcs in an extremal trajectory (see lemmas 4.2 and A.3). A circle on an axis describes an equilibrium point of the dynamics. The same diagram holds for $p$, but with all the arrow directions reversed.

Suppose that $x(0) = L/(L + \ell)$. Then considering the regular cases depicted in figure 5, we see that again $x(T) < L/(L + \ell)$, as $x(t)$ can increase towards $L/(L + \ell)$ only in the fourth case depicted in figure 5, yet it can never reach $L/(L + \ell)$, as this is an equilibrium (and thus an invariant set) of this dynamics.

Summarizing, we showed that $x(0) > \ell/(L + \ell)$. Using a similar argument shows that $x(0) \in (\frac{\ell}{L+\ell}, \frac{L}{L+\ell})$, and this implies that $x(t) \in (\ell/(L + \ell), L/(L + \ell))$ for all $t \in [0, T]$. Similar arguments can be used to prove the corresponding statement for $p$. ∎

The following lemma excludes certain transitions between arcs.

**Lemma A.6.** *Let $\mathscr{X}$ be an extremal trajectory. If there exists $\tau \in [0, T]$ such that $\varphi_0(\tau)\varphi_1(\tau) < 0$, then $\varphi_0(t)\varphi_1(t) < 0$ for all $t \in [\tau, T]$.*

*Proof.* Without loss of generality (w.l.o.g.), assume that $\varphi_0(\tau) < 0$ and $\varphi_1(\tau) > 0$. Hence, $\tau \in E_-^0 \cap E_+^1$. Since both sets are open, there exists a connected component $\mathcal{T} \subset E_-^0 \cap E_+^1$ such that $\tau \in \mathcal{T}$. Let $\mathcal{T}_0, \mathcal{T}_1$ be the connected components containing $\tau$ with respect to $E_-^0, E_+^1$, respectively. Then, $\mathcal{T} = \mathcal{T}_0 \cap \mathcal{T}_1$. Let $\tau_i, i = 1, \ldots, 4$, be such that $\mathcal{T}_1 = (\tau_1, \tau_3), \mathcal{T}_0 = (\tau_2, \tau_4)$. W.l.o.g., assume that $\tau_1 \le \tau_2$.

Assume first that $\tau_2 > 0$. Then there are three possibilities: $\tau_3 < \tau_4$, $\tau_3 > \tau_4$ and $\tau_3 = \tau_4$; see figure 6. By definition, $\varphi_0(\tau_2) = 0$, $\varphi_1(\tau_2) > 0$. Lemma 4.2 implies that $u_0(t) = \ell$, $u_1(t) = L$ for all $t \in \mathcal{T}$. By (A 4) and (A 5), both $\varphi_0$ and $\varphi_1$ are differentiable on $\mathcal{T}$, and the right-derivative $D_{\tau_2}^+\varphi_0$ exists. Since $\varphi_0(\tau_2) = 0$ and $\varphi_0(t) < 0$ on $t \in \mathcal{T}$, we have

$$\begin{aligned} 0 &\ge D_{\tau_2}^+\varphi_0 \\ &= u_1(\tau_2^+)(p(\tau_2) - (1 - x(\tau_2))) \\ &= L(p(\tau_2) + x(\tau_2) - 1). \end{aligned} \tag{A 10}$$

Recall that $\dot{x}(t) = \ell - (\ell + L)x(t)$, $\dot{p} = (\ell + L)p(t) - L$ for $t \in \mathcal{T}$ (see figure 5), so

$$\left.\begin{aligned} x(t) &= \left(x(\tau_2) - \frac{\ell}{\ell + L}\right)e^{-(\ell+L)(t-\tau_2)} + \frac{\ell}{\ell + L} < \left(x(\tau_2) - \frac{\ell}{\ell + L}\right)e^{(\ell+L)(t-\tau_2)} + \frac{\ell}{\ell + L} \\ \text{and} \qquad p(t) &= \left(p(\tau_2) - \frac{L}{\ell + L}\right)e^{(\ell+L)(t-\tau_2)} + \frac{L}{\ell + L}, \end{aligned}\right\} \tag{A 11}$$

where the inequality (A 11) follows from the fact that $x(t) > \frac{\ell}{\ell+L}$ (see lemma A.5). Summing up these equations gives

$$x(t) + p(t) < (x(\tau_2) + p(\tau_2) - 1)e^{(\ell+L)(t-\tau_2)} + 1.$$

Thus, for any $t \in \mathcal{T}$,

$$\begin{aligned} \dot{\varphi}_0(t) &= L(p(t) + x(t) - 1) \\ &< L(x(\tau_2) + p(\tau_2) - 1)\,e^{(\ell+L)(t-\tau_2)} \\ &\le 0, \end{aligned}$$

where the last inequality follows from (A 10). Hence, $\dot{\varphi}_0(t) < 0$ on $\mathcal{T}$. Since $\operatorname{sgn}\dot{\varphi}_1(t) = -\operatorname{sgn}\dot{\varphi}_0(t)$, $\dot{\varphi}_1(t) > 0$ for all $t \in \mathcal{T}$.

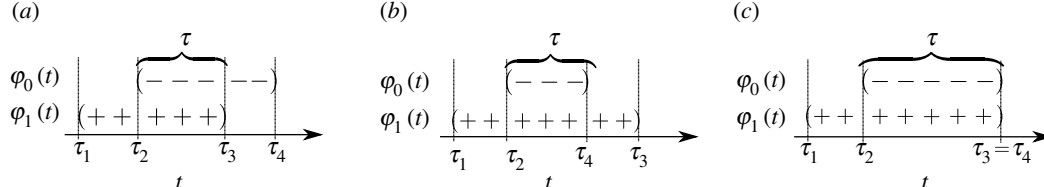

**Figure 6.** An illustration of the three cases studied in the proof of lemma A.6: (a) $\tau_3 < \tau_4$, (b) $\tau_3 > \tau_4$ and (c) $\tau_3 = \tau_4$.

We now show that $\tau_3 = \tau_4$. Assume that $\tau_3 < \tau_4$. Then, $\varphi_0(\tau_3) < 0$ and $\varphi_1(\tau_3) = 0$ as shown in figure 6a. Integrating $\dot{\varphi}_1$ over $\mathcal{T}$, and since $\varphi_1(\tau_2) \geq 0$, we get that $\varphi_1(\tau_3) > 0$, which is a contradiction. Similarly, assuming that $\tau_3 > \tau_4$ gives $\varphi_0(\tau_4) = 0$ (see figure 6b). Since $\varphi_1(\tau_2) = 0$ and $\dot{\varphi}_0(t) < 0$ on $\mathcal{T}$, then $\varphi_0(\tau_4) < 0$, which is a contradiction. Thus, $\tau_3 = \tau_4$ (see figure 6c).

Let $\tau_e := \tau_3 = \tau_4$. Then, the preceding argument shows that $\varphi_0(\tau_e) < 0$ and $\varphi_1(\tau_e) > 0$. If $\tau_e < T$, then the definition of $\tau_3$, $\tau_4$ implies that $\phi_0(\tau_e)\phi_1(\tau_e) = 0$. We conclude that $\tau_e = T$. i.e. $\sup \mathcal{T} = T$.

Assume now that $\inf \mathcal{T} = 0$. Since $\mathcal{X}$ is periodic, we can study it on the interval $[0, 2T]$. Let $\tilde{E}_-^0 := E_-^0 \cup (\{T\} + E_-^0)$, $\tilde{E}_+^1 := E_+^1 \cup (\{T\} + E_+^1)$. Hence, define $\tilde{\mathcal{T}}$ as the maximal open neighbourhood containing $\tau = T$ in $\tilde{E}_-^0 \cap \tilde{E}_+^1$. The sets $\tilde{\mathcal{T}}_1$, $\tilde{\mathcal{T}}_2$ are defined similarly. Replicating the previous arguments to the sets $\tilde{\mathcal{T}}$, $\tilde{\mathcal{T}}_1$, $\tilde{\mathcal{T}}_2$, we see that $\sup \tilde{\mathcal{T}} = 2T$. Hence, by periodicity, $\sup \mathcal{T} = T$. ∎

We can strengthen lemma A.6 to exclude mixed-sign arcs.

**Lemma A.7.** *Let $\mathcal{X}$ be an extremal trajectory. Then, $\varphi_0(t)\varphi_1(t) \geq 0$ for all $t \in [0, T]$.*

*Proof.* Assume that there exists $\tau \in [0, T]$ such that $\varphi_0(\tau)\varphi_1(\tau) < 0$. Lemma A.6 implies that $\varphi_0(t)\varphi_1(t) < 0$ for all $t \in [\tau, T]$. By periodicity, $\varphi_0(0)\varphi_1(0) < 0$. Applying lemma A.6 gives $\varphi_0(t)\varphi_1(t) < 0$ for all $t \in [0, T]$. This implies that both $\varphi_0(t)$, $\varphi_1(t)$ have constant and opposite signs. W.l.o.g., assume that $\varphi_0(t) < 0$ and $\varphi_1(t) > 0$ for all $t \in [0, T]$. By lemma 4.2, $u_0(t) \equiv \ell$ and $u_1(t) \equiv L$. Hence, $\dot{x}(t) = \ell - (\ell + L)x(t)$ for all $t \in [0, T]$, so

$$x(T) - x(0) = \left(\frac{\ell}{\ell + L} - x(0)\right)(1 - e^{-(\ell+L)T}).$$

Since $x(0) > \ell/(\ell + L)$ (see lemma A.5), $x(T) < x(0)$, and this is a contradiction. ∎

For an extremal trajectory $\mathcal{X}$, recall that $E_r := \{t \in [0, T] \mid \varphi_0(t)\varphi_1(t) \neq 0\}$. Lemma A.7 implies that

$$E_r = E_{++} \cup E_{--}, \tag{A 12}$$

where

$$E_{++} := E_+^0 \cap E_+^1 \quad \text{and} \quad E_{--} := E_-^0 \cap E_-^1.$$

In other words, the only possible bang arcs are the first two cases in figure 5. Note that $1/2$ is an equilibrium point of both these arcs. Also, on a singular arc $x(t)$ is constant. This proves the following.

**Lemma A.8.** *Let $\mathcal{X}$ be an extremal trajectory. If $x(\tau) \neq 1/2$ for some $\tau \in [0, T]$, then $x(t) \neq 1/2$ for all $t \in [0, T]$*

The next lemma shows that an extremal trajectory must consist of a single singular arc.

**Lemma A.9.** *Let $\mathcal{X}$ be an extremal trajectory. Then, $\varphi_0(t)\varphi_1(t) = 0$ for all $t \in [0, T]$. Furthermore, $x(t) \equiv x(0)$ for all $t \in [0, T]$.*

*Proof.* We consider two cases.

*Case 1.* Suppose that there exists a $\tau \in [0, T]$ such that $x(\tau) \neq 1/2$. We may assume w.l.o.g. that $x(\tau) > 1/2$. By lemma A.8, $x(t) > 1/2$ for all $t \in [0, T]$. Seeking a contradiction, assume that $\mu(E_r) > 0$. Then, (A 12) implies that $\dot{x}(t) < 0$ for all $t \in E_r$ (see figure 5). By lemma A.3, $\dot{x}(t) = 0$ for almost all $t \in E_s$. We conclude that $x(T) < x(0)$, and this is a contradiction. Thus, $\mu(E_r) = 0$.

*Case 2.* Suppose that $x(t) \equiv 1/2$. Then, lemma A.4 implies that $\mu(E_r) = 0$. ∎

We can now prove theorem 4.4. We already know that $\mathcal{X}$ consists of a single singular arc. Lemma A.3 implies that there exists a $c \in (0, 1)$ such that $x(t) \equiv c$ for all $t \in [0, T]$. Integrating (A 6) over $[0, T]$ yields $(1c - 1)\bar{u}_0 = \bar{u}_1$, and this completes the proof.

## A.4. An alternative proof of theorem 4.5

The alternative proof is inspired by the completing the square idea in [53], which was used to prove the result for a system with a controlled inflow and a constant outflow (proposed earlier by the authors in the preprint [54]). We show that a similar and simpler approach can be developed to tackle the more general case when both the inflow and outflow can be controlled independently of each other. The proof is based on two lemmas. Let $T > 0$. Recall that we use the notation $\bar{y} := \frac{1}{T} \int_0^T y(s)\, ds$.

**Lemma A.10.** *Let $x_1(t)$ be a solution of (4.11) satisfying $x_1(T) = x_1(0)$. Then, $\overline{u_0 x_1^k} = \overline{(u_0 + u_1)x_1^{k+1}}$ for any integer $k \geq 0$.*

*Proof.* Since $x_1^{k+1}(T) - x_1^{k+1}(0) = 0$, the integral of

$$\frac{1}{k+1}\frac{dx_1^{k+1}}{dt} = x_1^k \frac{dx_1}{dt} = x_1^k(u_0(1 - x_1) - u_1 x_1)$$

is zero, and the result follows. ∎

For $k = 0, 1$, lemma A.10 gives

$$\left.\begin{array}{r}\bar{u}_0 = \overline{(u_0 + u_1)x_1} \\[2mm] \overline{u_0 x_1} = \overline{(u_0 + u_1)x_1^2,}\end{array}\right\} \tag{A 13}$$

and

respectively.

**Lemma A.11.** *Let $x_1(t)$ be a solution of (4.11) satisfying $x_1(T) = x_1(0)$. Then,*

$$\overline{u_1 x_1} \leq z, \tag{A 14}$$

*where $z := \bar{u}_0 \bar{u}_1 / (\bar{u}_1 + \bar{u}_0)$.*

*Proof.* Writing $\overline{u_1 x_1} = \overline{((u_0 + u_1) - u_0)x_1}$, and using (A 13) gives $\overline{u_1 x_1} = \bar{u}_0 - \overline{u_0 x_1} = \bar{u}_0 - \overline{(u_0 + u_1)x_1^2}$. To apply a completion of squares argument, write this as $\overline{u_1 x_1} = z + (\bar{u}_0 \bar{u}_0/(\bar{u}_0 + \bar{u}_1)) - \overline{(u_0 + u_1)x_1^2}$. Now (A 13) gives

$$\overline{u_1 x_1} = z - \left(\frac{\bar{u}_0 \bar{u}_0}{\bar{u}_0 + \bar{u}_1} - \frac{2\bar{u}_0}{\bar{u}_0 + \bar{u}_1}\overline{(u_0 + u_1)x_1} + \overline{(u_0 + u_1)x_1^2}\right)$$

$$= z - \overline{(u_0 + u_1)\left(x_1 - \frac{\bar{u}_0}{\bar{u}_0 + \bar{u}_1}\right)^2},$$

and this completes the proof. ∎

Equation (A 14) implies that $x_1(t) \equiv \frac{\bar{u}_0}{\bar{u}_1 + \bar{u}_0}$ is an optimal trajectory, thus providing an alternative proof to theorem 4.4.

## A.5. Proof of proposition 4.11

The Hamiltonian is $\mathcal{H} = p_1(u_0(1 - x_1) - \lambda_1 x_1) + p_2 u_0 + \beta(t)\lambda_1 x_1$, where we assume w.l.o.g. that $p_0 = T$. Hence, (3.5) gives

$$\dot{p}_1(t) = (u_0(t) + \lambda_1)p_1(t) - \lambda_1 \beta(t)$$

and

$$\dot{p}_2(t) \equiv 0.$$

The switching function is

$$\varphi_0 = p_1(1 - x_1) + p_2(0), \tag{A 15}$$

and thus

$$\dot{\varphi}_0 = \lambda_1(p_1 - \beta + \beta x_1)$$

and
$$\ddot{\varphi}_0 = \lambda_1((u_0 + \lambda_1)p_1 - \lambda_1\beta - \dot{\beta} + \dot{\beta}x_1 + \beta(u_0(1 - x_1) - \lambda_1 x_1)). \tag{A 16}$$

By lemma 4.2, an optimal control is bang-bang on $E_r$. Hence, we study the control on the set $E_0 := \{t \mid \varphi_0(t) = 0\}$, which we assume to have non-zero measure. As in the proof of lemma A.3, we can find a set $F \subseteq E_0$ such that $F = E_0$ a.e. and $\varphi(t) = \dot{\varphi}(t) = \ddot{\varphi}(t)$ for all $t \in F$. This gives

$$p_1(t) = \beta(t)(1 - x_1(t)),$$

$$(1 - x(t))^2 = -p_2(0)/\beta(t)$$

and
$$u_0(t) = \frac{\lambda_1}{1 - x(t)} - \lambda_1 + \frac{\dot{\beta}(t)}{2\beta(t)}, \tag{A 17}$$

and this proves (4.21). Note that this implies that $p_2(0) < 0$. Furthermore, if $E_0 = [0, 1]$, then the equation $\int_0^1 u_0(t) \, dt = \bar{u}_0$ yields (4.22).

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
