## [Peer Review File · Royal Society Open Science]

Review History

RSOS-210878.R0 (Original submission)

Review form: Reviewer 1

Is the manuscript scientifically sound in its present form?

Yes

Are the interpretations and conclusions justified by the results?

Yes

Is the language acceptable?

Yes

Do you have any ethical concerns with this paper?

No

Have you any concerns about statistical analyses in this paper?

No

Recommendation?

Accept as is

Comments to the Author(s)

Thank you for addressing my and the other reviewers' previous comments.

One minor thing:

I believe line 46 should read 'either the minimum oR THE maximum'

Review form: Reviewer 2

Is the manuscript scientifically sound in its present form?

Yes

Are the interpretations and conclusions justified by the results?

Yes

Is the language acceptable?

Yes

Do you have any ethical concerns with this paper?

No

Have you any concerns about statistical analyses in this paper?

No

Recommendation?

Accept as is

Comments to the Author(s)

All three previous reports of the reviewers were overall very positive about the manuscript. They mostly mentioned minor issues or questions, while reviewer 2 and reviewer 3 had the following main reservations about the paper.

- 1) (Reviewer 2) The reviewer believed the mathematical work not to be clearly connected to the biology due to the fact that the models are too simplified.
- 2) (Reviewer 3 and Reviewer 2 to a lesser extent) The paper is not appropriate for this interdisciplinary journal as it is too theoretical/mathematical and too hard to follow.

I believe the authors have answered all questions and revised the manuscript in a convincing way. As for the two main concerns above, I do not agree with the reviewers. As the authors reply, the models are a simplification of the biological process, but there is indeed always a trade-off between very detailed models which are complex to analyze and simpler models with some approximations that allow for a more detailed analysis. I believe that the models used have a good balance between such simplicity and complexity, and certainly are still relevant biologically. Furthermore, while it is true that parts of the paper are very mathematical (and could therefore be difficult to follow for someone without such background), I do think that the motivation, introduction, and discussions in the work have been very clearly explained for researchers coming from different disciplines. The biological relevance and motivation is also convincing.

As all previous comments have been convincingly addressed, and as I personally also find the work very interesting, well written, and of high technical quality, I thus recommend the paper for publication in its current form.

Decision letter (RSOS-210878.R0)

Dear Dr Sontag,

I am pleased to inform you that your manuscript entitled "Maximizing average throughput in oscillatory biochemical synthesis systems: an optimal control approach" is now accepted for publication in Royal Society Open Science.

Please ensure that you send to the editorial office an editable version of your accepted manuscript, and individual files for each figure and table included in your manuscript. You can send these in a zip folder if more convenient. Failure to provide these files may delay the processing of your proof.

on behalf of Professor Professor R. Kerry Rowe (Subject Editor).

Associate Editor Comments to Author:

Thanks for the submission and our apologies that the decision has taken a little while to return to you - only one of the earlier Interface reviewers was available to assess your transferred paper. Nevertheless, this individual and a second reviewer each recommend acceptance as you have addressed any earlier comments/queries satisfactorily. There is one minor typographical tweak to take into account during the proofing stage of the manuscript (see reviewer one's comments), but otherwise good job and congratulations!

Reviewer(s)' Comments to Author:

Reviewer: 1

Comments to the Author(s)

Thank you for addressing my and the other reviewers' previous comments.

One minor thing:

I believe line 46 should read 'either the minimum or THE maximum'

Reviewer: 2

Comments to the Author(s)

All three previous reports of the reviewers were overall very positive about the manuscript. They mostly mentioned minor issues or questions, while reviewer 2 and reviewer 3 had the following main reservations about the paper.

1) (Reviewer 2) The reviewer believed the mathematical work not to be clearly connected to the biology due to the fact that the models are too simplified.

2) (Reviewer 3 and Reviewer 2 to a lesser extent) The paper is not appropriate for this interdisciplinary journal as it is too theoretical/mathematical and too hard to follow.

I believe the authors have answered all questions and revised the manuscript in a convincing way. As for the two main concerns above, I do not agree with the reviewers. As the authors reply, the models are a simplification of the biological process, but there is indeed always a trade-off between very detailed models which are complex to analyze and simpler models with some approximations that allow for a more detailed analysis. I believe that the models used have a good balance between such simplicity and complexity, and certainly are still relevant biologically. Furthermore, while it is true that parts of the paper are very mathematical (and could therefore be difficult to follow for someone without such background), I do think that the motivation, introduction, and discussions in the work have been very clearly explained for researchers coming from different disciplines. The biological relevance and motivation is also convincing.

As all previous comments have been convincingly addressed, and as I personally also find the work very interesting, well written, and of high technical quality, I thus recommend the paper for publication in its current form.
